# Simultaneous atmospheric water production and 24-hour power generation enabled by moisture-induced energy harvesting

Tingxian Li [1,2,3] ✉, Minqiang Wu[1,3], Jiaxing Xu [1,3], Ruxue Du[1], Taisen Yan[1], Pengfei Wang [1], Zhaoyuan Bai[1], Ruzhu Wang [1,2] & Siqi Wang[1]

Water and electricity scarcity are two global challenges, especially in arid and remote areas. Harnessing ubiquitous moisture and sunlight for water and power generation is a sustainable route to address these challenges. Herein, we report a moisture-induced energy harvesting strategy to realize efficient sorption-based atmospheric water harvesting (SAWH) and 24-hour thermo-electric power generation (TEPG) by synergistically utilizing moisture-induced sorption/desorption heats of SAWH, solar energy in the daytime and radiative cooling in the nighttime. Notably, the synergistic effects significantly improve all-day thermoelectric power density (~346%) and accelerate atmospheric water harvesting compared with conventional designs. We further demonstrate moisture-induced energy harvesting for a hybrid SAWH-TEPG device, exhibiting high water production of 750 g m$^{-2}$, together with impressive thermoelectric power density up to 685 mW m$^{-2}$ in the daytime and 21 mW m$^{-2}$ in the nighttime. Our work provides a promising approach to realizing sustainable water production and power generation at anytime and anywhere.

Freshwater shortage is one of the most urgent global challenges and ~2.2 billion people are suffering from insufficient drinking water[1]. Moreover, water shortage is becoming increasingly serious with population growth and environmental pollution, and more than half of the worldwide population will face clean water scarcity by the 2050 s[2]. People suffering from water shortage mostly live in land-locked regions and arid climates, where traditional freshwater collection technologies (e.g, rain collection, artificial precipitation, and desalination) are inaccessible to solve this challenge[3]. Alternatively, extracting fresh water from air is a possible way to get liquid drinkable water due to the huge moisture amount of 12,900 cubic kilometers, six times the total volume of rivers worldwide[4]. Several atmospheric water harvesting (AWH) technologies, including chiller-driven dew-water collection[5], radiative cooling-driven water condensation[6], hydrophilic surface-enabled fog harvesting[7], and sorption-based AWH (SAWH)[8], were reported and discussed. Among all AWH technologies, SAWH is

an exclusively efficient way to extract fresh water from air in arid climates at low relative humidity (RH), ascribing to its high affinity of sorbents with water vapor[9–11]. Considering the ubiquitous moisture and sufficient sunlight in arid regions, solar-driven SAWH is expected to provide safe drinking water for one billion people with 5 L day$^{-1}$ [12].

The feasibility of solar-driven SAWH in arid climates was clearly verified;[13] however, low water productivity is a long-standing challenge for solar-driven SAWH, almost an order of magnitude lower than that of solar-driven interfacial evaporation or desalination[14,15]. The poor performance not only can be ascribed to the low water sorption capacity of sorbents at low RH, but also as a result of the low thermal efficiency of solar-driven SAWH devices. In comparison with tradi-tional sorbents (e.g., silica gel and zeolites), the emerging metal-organic frameworks (MOFs) attract much attention in solar-driven SAWH due to higher water sorption capacity and lower regeneration temperature, wherein MOF-801[16,17], MOF-303[18], and $Co_2Cl_2(BTDD)$[19]

[1]Institute of Refrigeration and Cryogenics, School of Mechanical Engineering, Shanghai Jiao Tong University, Shanghai 200240, China. [2]Research Center of Solar Power and Refrigeration of Ministry of Education, Shanghai Jiao Tong University, Shanghai 200240, China. [3]These authors contributed equally: Tingxian Li, Minqiang Wu, Jiaxing Xu. ✉e-mail: Litx@sjtu.edu.cn

show distinct superiority over traditional sorbents. Recently, MOF-333 was reported as a promising SAWH material with high working capacity and low desorption temperature below 60 °C[20]. Besides MOFs, many low-cost sorption materials were also developed for realizing cost-effective solar-driven SAWH, such as hydrogels[21,22], ion solutions[23], and salt-based composites[24–27]. Nevertheless, the strong endothermic/exothermic effects of sorbents during moisture sorption/desorption need a large amount of energy consumption for addressing sorption/desorption heats, resulting in the low energy efficiency of SAWH devices[27]. To improve the overall performance of SAWH, it is highly desirable to develop efficient hybrid SAWH systems to utilize the endothermic/exothermic effects of sorbents for power generation[28], sorption heating[29], desorption cooling[30], etc.

In addition to freshwater shortage, electricity scarcity is another disadvantage in arid and remote areas. Due to the rapid consumption of fossil fuels and the presence of energy crisis, it is an urgent need to generate sustainable power to deal with climate change and growing global population[31,32]. Recently, solar-driven hybrid energy systems have been proposed for freshwater production via thermal-induced seawater evaporation or polluted water distillation and power generation via photovoltaic panels or salinity gradient[33–41]. However, these co-generation strategies are unworkable in arid and remote areas due to long-term water shortages. Solar-driven SAWH coupled with a thermoelectric generator can overcome the above drawbacks and realize freshwater and power co-generation via cascade utilization of solar thermal energy[28]. However, the natural intermittence of sunlight makes the reported solar co-generation systems impossible to realize nighttime power generation without energy input[39]. Fortunately, radiative cooling enables the thermoelectric generator to show the possibility of generating electricity during nighttime without additional energy consumption[42]. However, due to the small temperature difference between radiative cooling and ambient temperature, the thermoelectric power density is scarcely sufficient for typical applications of lighting, off-grid sensors, and digital communication[43].

Herein, we propose a moisture-induced energy harvesting strategy for realizing simultaneous sorption-based atmospheric water harvesting (SAWH) and 24-hour thermoelectric power generation (TEPG). We demonstrate a hybrid SAWH-TEPG device for water and power co-generation by subtly harnessing the synergistic thermal effects of moisture sorption/desorption, radiative heating from sunlight, and radiative cooling from the universe. In the daytime, solar energy drives the hybrid device to achieve atmospheric water production and thermoelectric power generation. The endothermic effect of the SAWH module during the moisture desorption of sorbent lowers the heat sink temperature of the TEPG module. As a result, the synergistic effects of desorption cooling from sorbents and radiative heating from sunlight enlarge the driving temperature difference and thus improve thermoelectric power density. In the nighttime, sorbents capture water vapor from air and release a large amount of sorption heat. The exothermic effect of the SAWH module during the moisture sorption of sorbent increases the hot-side temperature of the TEPG module; simultaneously, the coldness from the universe by radiative cooling lowers the cold-side temperature of the TEPG module. Moreover, the heat consumed by the TEPG module also accelerates the moisture sorption of the SAWH module by lowering sorbent temperature. Therefore, the synergistic effects not only improve thermoelectric power density but also accelerate atmospheric water harvesting. Importantly, both daytime and nighttime synergistic effects enable the hybrid SAWH-TEPG device to exhibit superior water production of 750 g m$^{-2}$ and impressive thermoelectric power density up to 685 mW m$^{-2}$ in the daytime and 21 mW m$^{-2}$ in the nighttime. Our work provides a potential route to realize efficient water production and 24-hour power generation at anytime and anywhere.

## Results

### Principle of moisture-induced energy harvesting for water and power generation

Moisture and sunlight are ubiquitous in nature anywhere, even in arid deserts and remote areas. We propose a moisture-induced energy harvesting strategy for sustainable water production and power generation by synergistically utilizing the thermal effects of hybrid SAWH-TEPG (Fig. 1). The hybrid device mainly consists of a dual-functional coating layer, a TEPG module, a SAWH module with packed sorbent, and an air-cooling condenser. The SAWH module is attached to the bottom surface of the TEPG module by using an aluminum block as a thermal transport framework. To suppress the energy loss during daytime and nighttime, the device is covered with a thermal insulation layer and polyethylene membrane. The working principle of this moisture-induced energy harvesting for water and power co-generation is as follows:

During daytime (Fig. 1a), the dual-functional coating layer works as a light-to-heat absorber for solar thermal collection on the top surface of the TEPG module. The temperature difference between the top hot-side and bottom cold-side drives the TEPG module to realize heat-to-electricity conversion via the Seebeck effect[40,44]. In conventional TEPG design, the heat released by the cold side of the TEPG module during power generation is directly dissipated to the ambient as waste heat. Differently, in our hybrid SAWH-TEPG device, the waste heat released by the TEPG module is reutilized to drive the SAWH module to realize moisture desorption for water production in air-cooling condenser. Moreover, moisture-desorption cooling lowers the cold-side temperature of the TEPG module by removing a large amount of heat through the desorption process of sorbent.

During nighttime (Fig. 1b), the dual-functional coating layer works as an infrared thermal emitter for radiative cooling on the top surface of the TEPG module. The SAWH module with packed sorbent is exposed to the ambient to capture water vapor from air. In conventional SAWH design, the sorption heat released by sorbent during the moisture sorption process is dissipated to the ambient directly as waste heat. However, in our hybrid SAWH-TEPG device, the sorption heat released by sorbent is utilized as a heat source for the hot side of the TEPG module, while the coldness obtained by radiative cooling from the universe is used as a heat sink for the cold side of TEPG module. Thus, the temperature difference between radiative cooling and sorption heating enables the TEPG module to realize power generation in the nighttime. It is worthwhile to note that the moisture-sorption heat of sorbent increases the hot-side temperature of the TEPG module, and the heat consumed by the TEPG module in turn further accelerates water vapor capture of the SAWH module from air by lowering the sorbent temperature.

We analyze the energy flux and heat transfer network of the hybrid SAWH-TEPG device during the water and power co-generation process in the daytime (Fig. 1c) as well as power generation and water vapor capture at nighttime (Fig. 1d). The detailed information of energy balance can be seen in Supplementary Note 1, Supplementary Note 2, and Supplementary Figure 1. For SAWH mode, the sorbents capture moisture from the air at nighttime and release water in the daytime to realize atmospheric water harvesting. For TEPG mode, the heat flux from its hot side to the cold side ($Q_{TE}$) for power generation can be expressed as:

$$Q_{TE} = A \frac{(T_{hot} - T_{cold})}{R_{TE}} = A \frac{\Delta T_{TE}}{R_{TE}} \qquad (1)$$

Where $A$ is the area of the TEPG module, $T_{hot}$ and $T_{cold}$ are the hot-side and cold-side temperatures of the TEPG module, $R_{TE}$ is the thermal resistance of the TEPG module, $\Delta T_{TE}$ is the temperature difference ($\Delta T_{TE} = T_{hot} - T_{cold}$) between the hot-side and cold-side of the TEPG

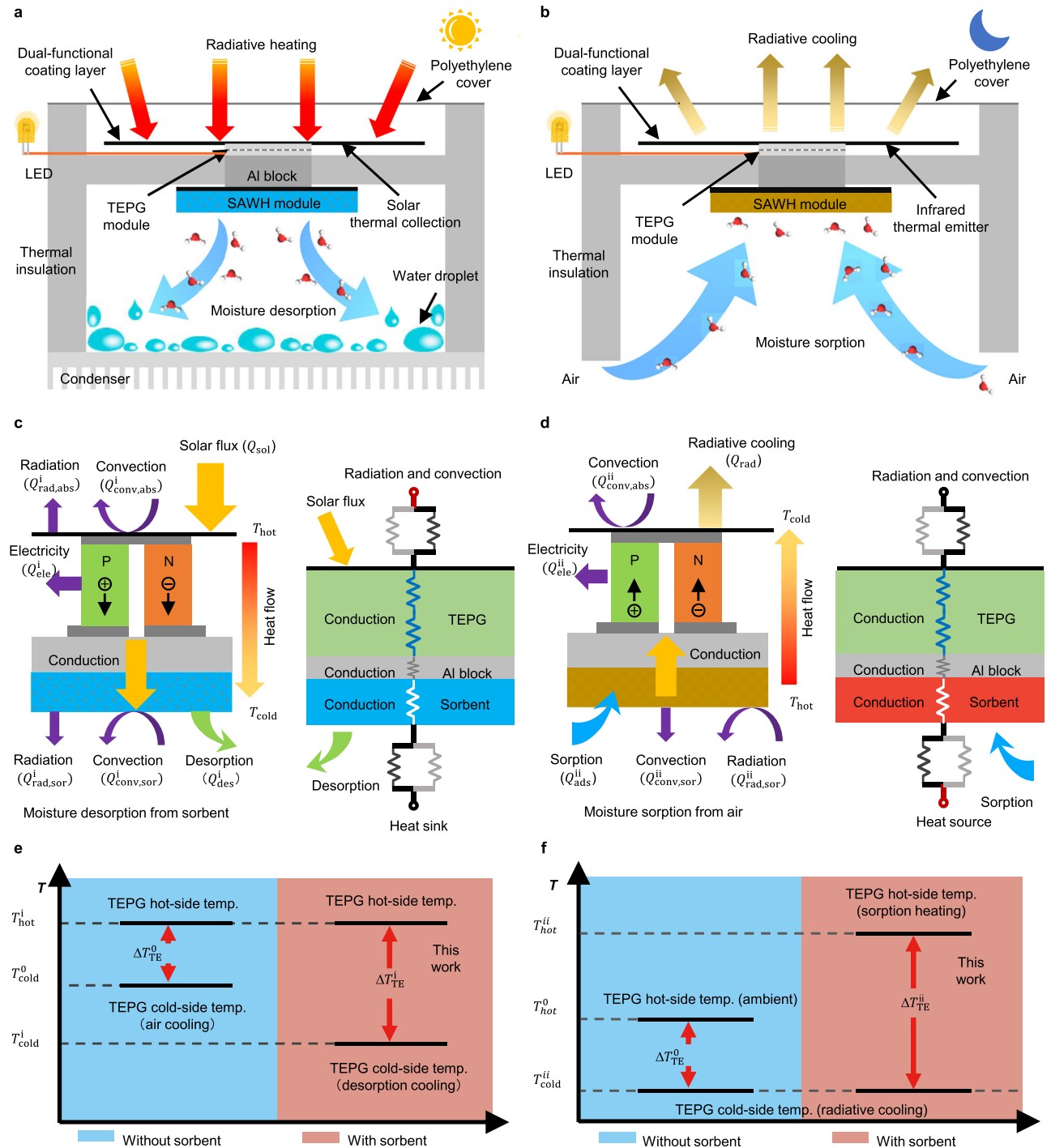

**Fig. 1 | Synergistic effects of moisture-induced energy harvesting for water and power co-generation. a** Schematic of water production and power generation by radiative heating from sunlight during daytime. **b** Schematic of water vapor capture from air and power generation by radiative cooling from the universe during nighttime. **c** Energy balance and thermal network of hybrid SAWH-TEPG by radiative heating during daytime. **d** Energy balance and thermal network of hybrid SAWH-TEPG by radiative cooling during nighttime. **e** The enlarged temperature difference for power generation by moisture-desorption-induced synergistic effect for lowering the cold-side temperature of the TEPG module during daytime. **f** The enlarged temperature difference for power generation by moisture-sorption-induced synergistic effect for increasing the hot-side temperature of TEPG module during nighttime.

module for power generation. The maximum thermoelectric power density ($P_{max}$) can be calculated as:

$$P_{max} = \frac{\left(n\left(S_{pn}\right)\Delta T_{TE}\right)^2}{4R_L} \qquad (2)$$

where $n$ is the number of TEPG modules, $R_L$ is the loading resistance, $S_{pn}$ is the Seebeck coefficient of thermoelectric material. In addition to the intrinsic merits of thermoelectric material and $R_L$, the theoretical maximum thermoelectric power density ($P_{max}$) is solely determined by the temperature difference of the TEPG module ($\Delta T_{TE}$). Therefore, the higher the temperature difference of the TEPG module, the higher the

thermoelectric power density of the hybrid SAWH-TEPG device is generated.

For the hybrid SAWH-TEPG in the daytime, the desorption heat consumed by the SAWH module during the moisture desorption of sorbent can lower the cold-side temperature of the TEPG module from $T_{cold}^0$ to $T_{cold}^i$ (Fig. 1e) compared with conventional air cooling-based TEPG design. Moreover, sorbent desorption cooling is usually two orders of magnitude higher than air-convective cooling[30]. As a result, the temperature difference of the TEPG module is enlarged from $\triangle T_{TE}^0$ to $\triangle T_{TE}^i$ (Fig. 1e), and the high $\triangle T_{TE}^i$ improves the thermoelectric power density of the TEPG module.

For the hybrid SAWH-TEPG in the nighttime, the sorption heat released by the SAWH module during the moisture sorption of sorbent increases the hot-side temperature of the TEPG module from $T_{hot}^0$ to $T_{hot}^{ii}$ (Fig. 1f) compared with the reported radiative cooling-based TEPG design[42]. Thus, the synergistic effects of sorption heating and radiative cooling enlarge the temperature difference of the TEPG module from $\triangle T_{TE}^0$ to $\triangle T_{TE}^{ii}$ (Fig. 1f). Furthermore, the heat consumed by the TEPG module also lowers the sorbent temperature and thus increases its local RH[45]. As a result, both low sorbent temperature and high local RH accelerate the moisture sorption of SAWH, thus making a significant contribution to improving water harvesting compared with conventional air-cooling design.

The proposed moisture-induced synergistic thermal effects, for the first time to our knowledge, not only improve the power density of the TEPG module and accelerate the water vapor capture of SAWH but also realize advanced 24-hour power generation without energy storage compared with conventional hybrid water and power co-generation systems[33–41].

## Synthesis and characterizations of sorbents for SAWH

Water sorption MOFs have been widely reported and show the advantages of high specific surface area, abundant pore structure, and typical S-shaped isotherm[46–48]. We selected MIL-101 (Cr)[49] as SAWH sorbent due to its high water uptake, fast water sorption kinetics, and low desorption temperature compared with other MOFs (Supplementary Figure 2). To accelerate the moisture sorption/desorption of MIL-101 (Cr) by overcoming its poor heat and mass transfer, we coat MIL-101(Cr) powders on a copper foam (CF) via the in-situ impregnation method to fabricate MIL-101(Cr)@CF composite as water sorption sorbent (Supplementary Note 3), and further characterize the morphology, sorption performance and thermal stability of MIL-101(Cr) @CF (Fig. 2, Supplementary Note 4).

Digital photographs and SEM images show the MIL-101(Cr) layer remains intact crystal structure after the coating procedure and uniformly distributes on the porous skeleton of CF (Fig. 2a). The moisture sorption-desorption isotherm reveals that MIL-101(Cr) has impressive water uptake (-1.2 $g_{water}\,g^{-1}_{MOF}$) (Fig. 2b, Supplementary Note 5) and temperature-insensitive sorption characteristics under different temperatures (Supplementary Figure 3), indicating the flexible adaptability of MIL-101(Cr)@CF for SAWH in winter or summer. $N_2$ adsorption isotherms show the Brunauer-Emmett-Teller (BET) surface area as high as 3224.6 $m^2\,g^{-1}$ (Supplementary Figure 4), together with a desirable fast water sorption kinetics of MIL-101(Cr)@CF (Supplementary Figure 5). Moreover, TGA-DSC shows the thermal energy density of MIL-101(Cr) during moisture sorption-desorption is up to 2500 kJ kg$^{-1}$ (Fig. 2c), approximately 10 times higher than that of conventional solid-liquid[50] or solid-solid[51] phase change materials. The high thermal density can be used to drive the hybrid SAWH-TEPG device to realize power generation.

The thermal conductivity of sorbent plays a key role in accelerating thermal transport during the moisture sorption/desorption process. Compared with MIL-101(Cr) powders, MIL-101(Cr)@CF composite exhibits high thermal conductivity (3.54 W m$^{-1}$ K$^{-1}$) (Fig. 2d) and enhanced sorption kinetics under different RH conditions (Figures S5).

Specifically, the thermal conductivity of MIL-101(Cr)@CF composite is approximately 28 times higher than that of MIL-101(Cr) powders (0.12 W m$^{-1}$ K$^{-1}$). These results indicate the porous structure and high thermal conductivity of MIL-101(Cr)@CF composite can provide excellent water vapor capture and thermal transport ability during the moisture sorption/desorption process. Due to the introduction of powder binder into MIL-101(Cr)@CF, its water uptake (-0.93 $g_{water}\,g^{-1}_{MOF}$) is slightly lower than the pristine MOF (Supplementary Figure 6). MIL-101(Cr)@CF composite always has a higher temperature than the ambient temperature during the moisture sorption process due to a large amount of sorption heat released by the MOF sorbent (Supplementary Figure 7). The maximum temperature difference between composite and ambient is up to 10 °C (Supplementary Figure 8), and the high-temperature difference is utilized to improve the thermoelectric power density of the hybrid SAWH-TEPG device. Moreover, the MIL-101(Cr)@CF composite also exhibits excellent cyclic stability (Fig. 2e), mechanical strength (Supplementary Figure 9), and chemical stability (Supplementary Figs. 10 and 11) during repeated moisture sorption-desorption cycles.

Aiming at realizing both daytime radiative heating and nighttime radiative cooling for water and power co-generation, we design a simple and low-cost dual-functional coating layer for the hybrid SAWH-TEPG device by covering black paint on a thin copper plate (100 × 100 × 0.8 mm) (Fig. 2f). The coating layer shows a high solar absorption of ~95% and a high emissivity of ~90% during the wavelength range of 8–13 μm, indicating the dual-functional coating layer can efficiently work as a light-to-heat absorber for radiative heating in the daytime or thermal emitter for radiative cooling in the nighttime. Moreover, the working modes of the dual-functional coating layer can be automatically switched between absorber and emitter owing to its blackbody radiation characteristics.

## Proof-of-concept of hybrid SAWH-TEPG device

We design a pioneering hybrid SAWH-TEPG device to perform the proof-of-concept experiments of moisture-induced energy harvesting for water and power co-generation in our laboratory (Supplementary Note 6). The hybrid SAWH-TEPG device is assembled by a dual-functional coating layer, TEPG module, and MIL-101(Cr)@CF with a layer-by-layer structure (Fig. 3a). To verify the moisture-induced synergistic effects for promoting water vapor capture and power generation, we specially design a referenced device without MIL-101(Cr) compared to the hybrid SAWH-TEPG device with MIL-101(Cr)@CF (Supplementary Figure 12). The open-circuit voltage ($V_{oc}$) and short-circuit current ($I_{sc}$) of the TEPG module are investigated by the current-voltage (I-V) sweep under a temperature difference of 2 °C between its hot-side and cold-side ($\Delta T_{TE} = T_{hot} - T_{cold}$) (Supplementary Figure 13). Meanwhile, a water-cooling heat sink with 5 °C lower than the ambient is employed to simulate the radiative cooling effect in the laboratory by referring to the reported results[42,52,53]. The detailed experiments are described in Supplementary Note 7.

To evaluate the hybrid SAWH-TEPG device during nighttime under simulated conditions, we carried out indoor experiments of water vapor capture and power generation inside an environment-controlled chamber with 65% RH and 25 °C (Supplementary Figure 14). The MIL-101(Cr)@CF composite captures water vapor from air and its mass gradually increases during the moisture sorption process (Fig. 3b). The sorption heat released by MIL-101(Cr) makes the composite temperature increase sharply at the initial stage and then become stable along with moisture sorption. In contrast with the referenced device, this hybrid SAWH-TEPG device exhibits the highest temperature difference $\Delta T_{TE}$ for thermoelectric power generation by combining moisture-induced sorption heating with radiative cooling (Fig. 3c). The maximum voltages of the TEPG module are 115.8, 62.1, and 20.7 mV, respectively, for the devices based on sorption heating-

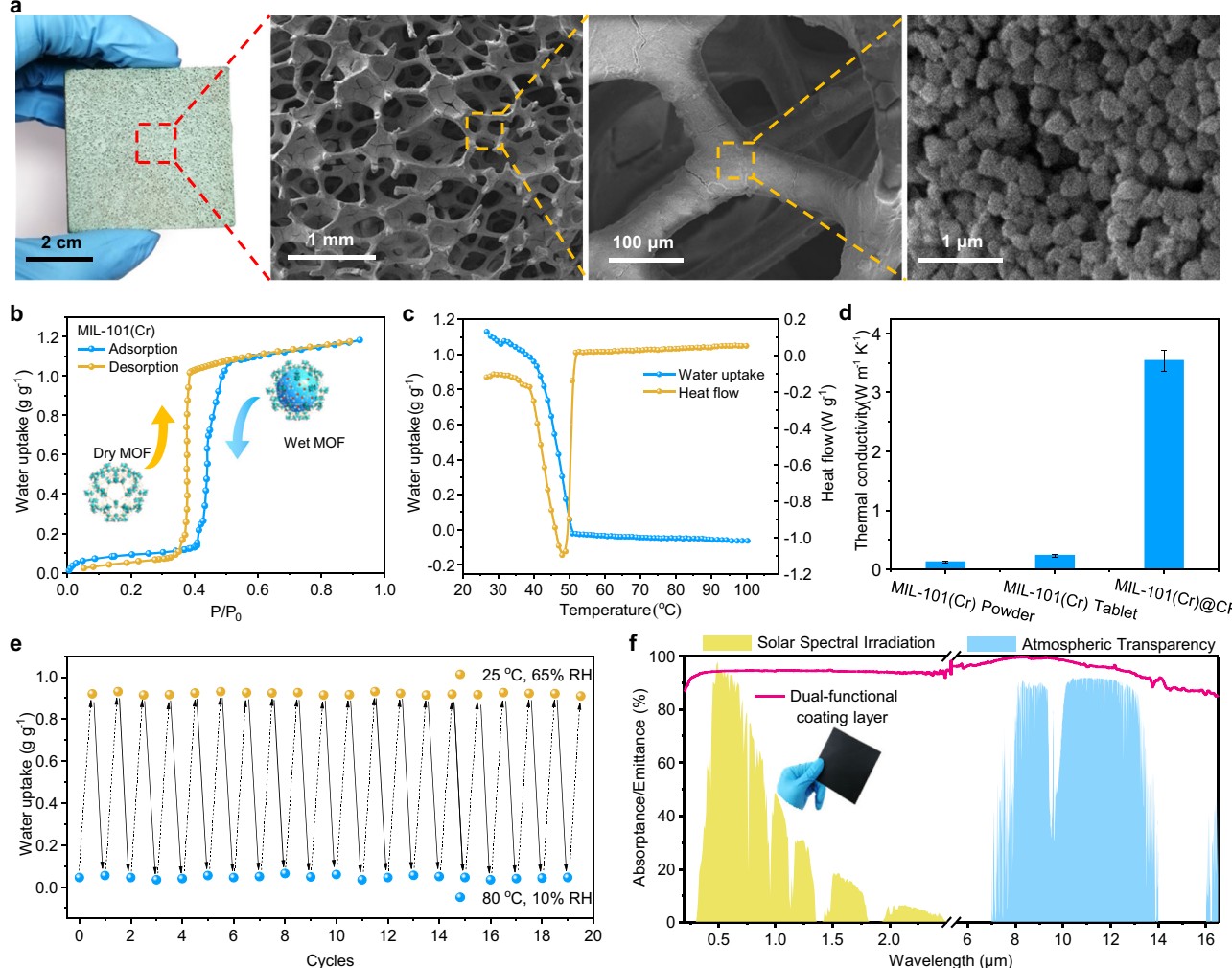

**Fig. 2 | Synthesis and characterization of MIL-101(Cr)@CF composite for water sorption. a** Digital photograph and SEM images of MIL-101(Cr)@CF composite (size: 50 × 50 × 5 mm). **b** Moisture sorption-desorption isotherms of MIL-101(Cr) with typical S-shaped characteristics and high water uptake at 25 °C. **c** Thermogravimetric analysis coupled with differential scanning calorimetry (TGA-DSC) of MIL-101(Cr) during the moisture desorption process. **d** Thermal conductivities of MIL-101(Cr) powder, MIL-101(Cr) tablet, and MIL-101(Cr)@CF composite. Error bar: standard deviation (SD). **e** Water uptake stability of MIL-101@CF composite during repeated moisture sorption-desorption cycles under 25 °C at 65% RH for sorption and 80 °C at 10% RH for desorption. **f** Solar absorbance and thermal emittance of the dual-functional coating layer under different wavelengths of radiation.

radiative cooling, radiative cooling, and sorption heating (Fig. 3d); and the corresponding maximum $P_{max}$ are 42.6, 12.3, and 1.6 mW m$^{-2}$ (Supplementary Figure 15). Therefore, synergistic effects of sorption heating and radiative cooling in the hybrid SAWH-TEPG device improve the thermoelectric power density by 346% compared with conventional radiative cooling-based TEPG design.

To evaluate the hybrid SAWH-TEPG device during daytime under simulated conditions, we carried out indoor experiments using a solar simulator with different sunlight intensities (Supplementary Figure 16). The solar simulator not only drives moisture desorption but also triggers thermoelectric power generation (Fig. 4a). The TEPG modules in the hybrid SAWH-TEPG device with MIL-101(Cr)@CF and the referenced device without MIL-101(Cr) have similar temperature evolutions at different simulated sunlight intensities (Fig. 4b), and the TEPG bottom temperature increases with increasing sunlight intensity, indicating the sunlight can be effectively converted into heat and then transferred to MIL-101(Cr)@CF. However, the former has a lower bottom temperature than the latter owing to the strong endothermic effect of MIL-101(Cr)@CF during moisture desorption. For instance, under one standard sun (1000 W m$^{-2}$), the maximum TEPG bottom temperature of the hybrid SAWH-TEPG device with MIL-101(Cr)@CF is 63.4 °C, -9.3 °C lower than that of the referenced device without MIL-101(Cr).

The hybrid SAWH-TEPG device and the referenced device have similar evolutions of open-circuit voltages ($V_{oc}$) at different simulated sunlight intensities (Fig. 4c). For a given TEPG module, its open-circuit voltage depends on the temperature difference between its hot side and cold side. A high sunlight intensity can generate a high hot-side temperature, and thus produce high $V_{oc}$ and thermoelectric power density. As a result, the maximum $V_{oc}$ increases with increasing sunlight intensity. For example, the maximum $V_{oc}$ are 383.3, 553.8, and 729.4 mV under the solar flux of 500, 750, and 1000 W m$^{-2}$; and the corresponding $P_{max}$ are 467.3, 968.3, and 1726 mW m$^{-2}$, respectively (Supplementary Figure 17). Furthermore, the SAWH-TEPG device has a higher voltage than the referenced device for a given sunlight intensity (Fig. 4c). To be more specific, under one standard sun (1000 W m$^{-2}$), the average $V_{oc}$ is improved from 492.3 to 571.9 mV by using MIL-101(Cr)@CF. This is because the former has a higher temperature difference for thermoelectric power generation due to its lower cold-side temperature contributed by the heat consumption of MIL-101(Cr)@CF during moisture desorption (Fig. 4b). In addition, this hybrid device exhibits accelerated water

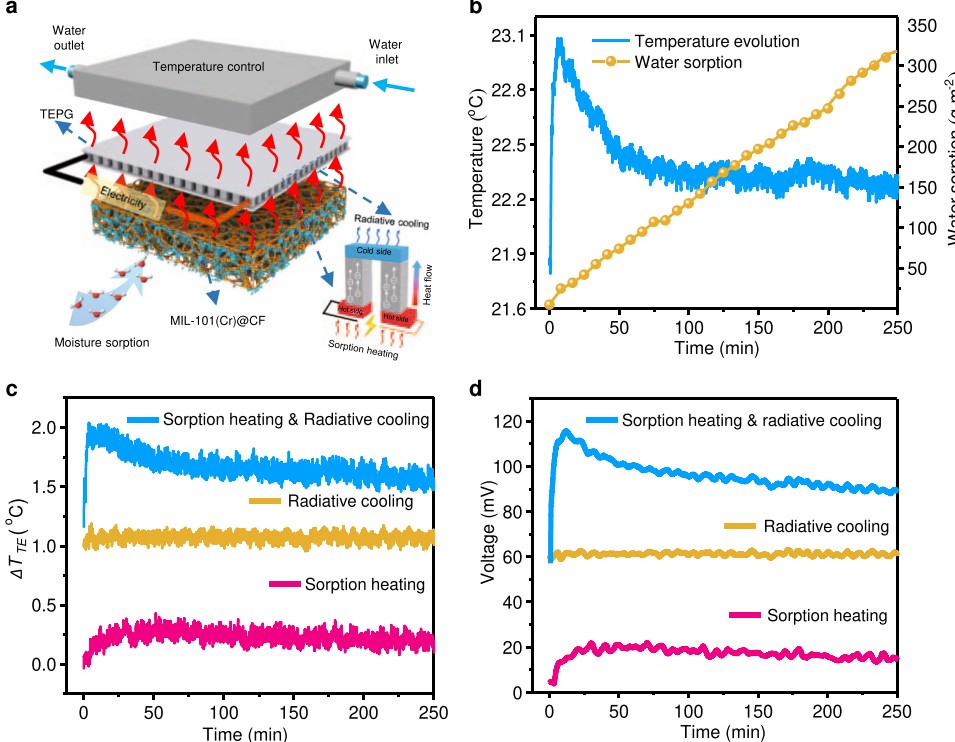

**Fig. 3 | Demonstration of synergistic effect of hybrid SAWH-TEPG device in nighttime.** All experiments were conducted at a typical ambient condition of 25 °C and 65% RH. **a** Schematic diagram of hybrid SAWH-TEPG device with MIL-101(Cr)@CF using a simulated radiative cooling source. **b** Temperature evolution and water sorption of MIL-101(Cr)@CF in the hybrid SAWH-TEPG device during moisture sorption process. **c** Temperature difference of the TEPG module under combined sorption heating & radiative cooling, radiative cooling, and sorption heating. **d** Voltage evolutions of the TEPG module under combined sorption heating & radiative cooling, radiative cooling, and sorption heating.

productivity of 150 g m$^{-2}$ h$^{-1}$ and high sunlight intensity accelerates water desorption rates (Fig. 4d). These results demonstrate that the hybrid SAWH-TEPG device has excellent water and power co-generation, and synergistic effects of SAWH and TEPG improve the open-circuit voltage by 120% compared with that of conventional air cooling-based TEPG design.

## Outdoor demonstration of water production and 24-hour power generation

We further carried out outdoor experiments on the roof of our Green Energy Lab (GEL) in Shanghai, China (Fig. 5a), the right one is the hybrid SAWH-TEPG device with MIL-101(Cr)@CF and the left one is the referenced device without MIL-101(Cr). During the nighttime, the hybrid SAWH-TEPG device captures water vapor from air and generates power by synergistically harvesting the coldness from the universe by radiative cooling and the sorption heating from MIL-101(Cr)@CF during moisture sorption. The results (Fig. 5b) clearly show that the MOF has the highest temperature due to its exothermic effect during moisture sorption, and the moisture-induced sorption heat makes the bottom temperature of the TEPG module higher than its top temperature. As a result, the temperature difference drives the TEPG module to realize heat-to-electricity conversion with open-circuit voltage $V_{oc}$ as high as 80 mV.

Moreover, contrary to the conventional radiative cooling-based TEPGs with cold-side temperature below ambient temperature[42,43], this hybrid SAWH-TEPG device has a cold-side temperature (TEPG top temperature) above ambient temperature although the top is cooled by radiative cooling (Fig. 5b). These results indicate the sorption heating from the MOF sorbent is higher than the radiative cooling from the universe. Moreover, we found the radiative cooling power becomes higher when compared with the reference device enabled by

the sorption heating of the MOF because its higher cold-side temperature enlarges the temperature difference for radiative heat transfer between TEPG and the universe (Supplementary Figure 18). We further analyze radiative cooling power and sorption heating power during the nighttime (Supplementary Figure 19, Supplementary Note 2), showing radiative cooling power is up to 90 W m$^{-2}$, and sorption heating power by the MOF is up to 130 W m$^{-2}$ (Supplementary Figure 20). The combination of sorption heating and radiative cooling in the hybrid energy device further enlarges the temperature difference $\Delta T_{TE}$ for thermoelectric power generation compared with the referenced device (Fig. 5c). The former has a higher $V_{oc}$ (~80 mV) than the latter (~60 mV), and the resultant maximum $P_{max}$ can be largely enhanced by 200% (Supplementary Figure 21). The outdoor experiments demonstrate that thermoelectric performance can be significantly enhanced by synergistic effects of sorption heating and radiative cooling, confirming that our moisture-induced synergistic energy harvesting is impressively more effective to realize power generation during nighttime.

During the daytime, the hybrid SAWH-TEPG device realizes water and power co-generation by synergistically harvesting radiative heating from sunlight and desorption cooling from moisture desorption (Fig. 5d). This hybrid device is sealed after moisture sorption overnight and then exposed to natural sunlight to perform moisture desorption and water collection. Solar irradiation converts into heat and induces the top temperature of the TEPG module higher than its bottom temperature, and the resultant temperature difference drives the TEPG module to realize heat-to-electricity conversion (Supplementary Figure 22). The maximum $V_{oc}$ is as high as 433.7 mV when solar irradiation reaches its peak value and the corresponding maximum $P_{max}$ is 685 mW m$^{-2}$ (Supplementary Figure 23). Simultaneously, the heat released by the bottom cold-side of the TEPG module drives

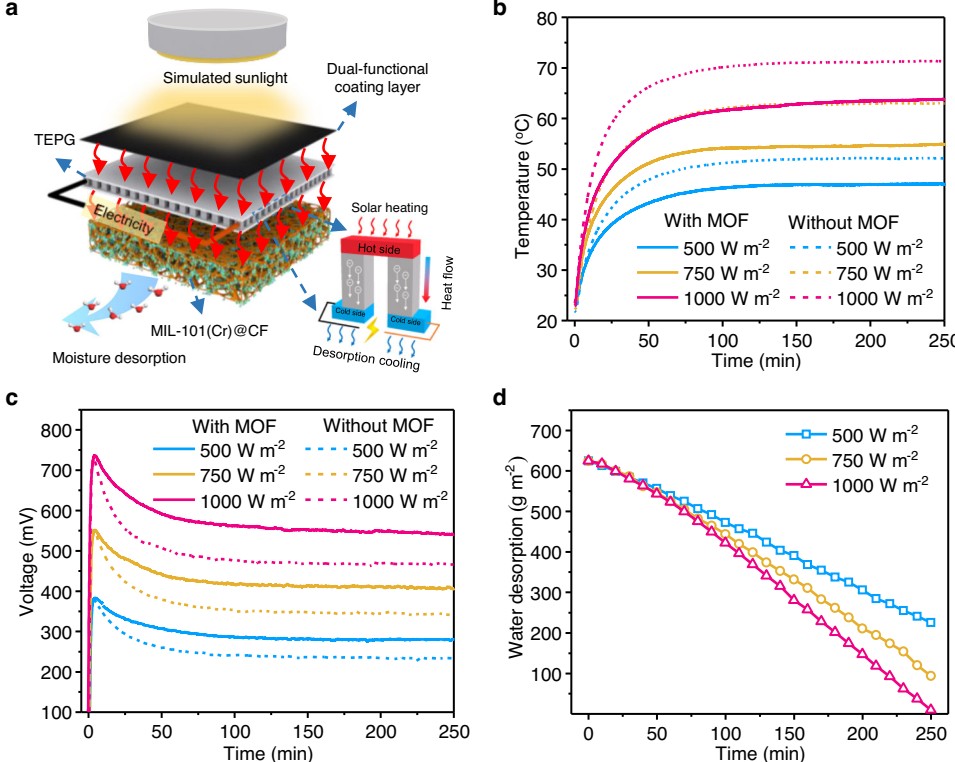

**Fig. 4 | Demonstration of synergistic effect of the hybrid SAWH-TEPG device in the daytime. a** Schematic diagram of the moisture desorption and power generation under the solar simulator. **b** Temperature evaluations of TEPG bottom in the hybrid SAWH-TEPG device with MIL-101(Cr)@CF and the referenced device without MIL-101(Cr). **c** Voltage evaluations of the TEPG module in the hybrid device with MIL-101(Cr)@CF and the referenced device without MIL-101(Cr) under different simulated sunlight intensities. **d** Water desorption of the hybrid SAWH-TEPG device under different simulated sunlight intensities. All experiments were conducted at an ambient temperature of 20 °C.

MIL-101(Cr)@CF to realize moisture desorption, and the desorbed water vapor removes heat from the TEPG module and thus enlarges the temperature difference for thermoelectric power generation. The released water vapor condensates on the surface of an air-cooled aluminum condenser with natural convective heat transfer (Supplementary Figure 25), and then becomes water droplets to be collected (Fig. 5e). The outdoor experiment reveals that the desorption cooling power is up to 300 W m$^{-2}$ (Supplementary Figure 24), and the solar-driven water harvesting efficiency ($\eta$) is calculated as 21.7% by the following equation,

$$\eta = \frac{m_{water} \triangle H_{water}}{\int_{t_0}^{t_{end}} W_{solar} dt} \quad (3)$$

where the $m_{water}$ represents the mass of harvested water, the $\triangle H_{water}$ is the enthalpy (latent heat) of water condensation, and the $W_{solar}$ is the real-time solar irradiation intensity during the water production process. The hybrid SAWH-TEPG device exhibits stable water collection performance during repeated outdoor experiments (Fig. 5f), confirming the hybrid device has excellent cyclic performance. We further predict the annual water harvesting of the proposed SAWH-TEPG device under the climate data of six typical different cities (Supplementary Figure 26), showing the annual water production is 295.7–612.6 L m$^{-2}$ a$^{-1}$ under different climate conditions (Supplementary Figure 27).

Furthermore, we carried out continuous week-long outdoor experiments to demonstrate the simultaneous atmospheric water production and 24-hour power generation (Supplementary Figure 28). The hybrid SAWH-TEPG device exhibits continuous voltage output with a maximum open-circuit voltage of 505 mV and stable water harvesting performance with average water uptake of 800 g m$^{-2}$.

Notably, although the hybrid SAWH-TEPG device achieves almost complete sorption and desorption of water vapor, there is still a long equilibration duration between the two processes, suggesting that the atmospheric water production and the power generation performance can be further improved by optimizing sorbent and device structure.

## Discussion

In summary, we propose a moisture-induced energy harvesting strategy for realizing energy-efficient atmospheric water production and 24-hour thermoelectric power generation and further demonstrate a hybrid SAWH-TEPG device for water and power co-generation by subtly harnessing the exothermic/endothermic effects of moisture sorption/desorption, radiative heating from sunlight, and radiative cooling from the universe. During the nighttime, the sorption heat released by SAWH during moisture sorption increases the hot side temperature of TEPG, while the coldness by radiative cooling lowers the cold side temperature of TEPG. The synergistic effects between sorption heating and radiative cooling enhance thermoelectric power density by as high as 346% compared with that of conventional design. Moreover, heat consumption by TEPG in turn accelerates water vapor capture from the air for SAWH. During the daytime, the desorption heat consumed by SAWH during moisture desorption lowers the cold side temperature of TEPG, and the synergistic effects between desorption cooling and radiative heating enlarge the temperature difference and thus improve thermoelectric power density. More importantly, both daytime and nighttime synergistic effects enable the hybrid SAWH-TEPG device to exhibit high water production of 750 g m$^{-2}$ and impressive all-day thermoelectric generation up to 685 mW m$^{-2}$ in the daytime and 21 mW m$^{-2}$ in the nighttime, achieving one of highest power density

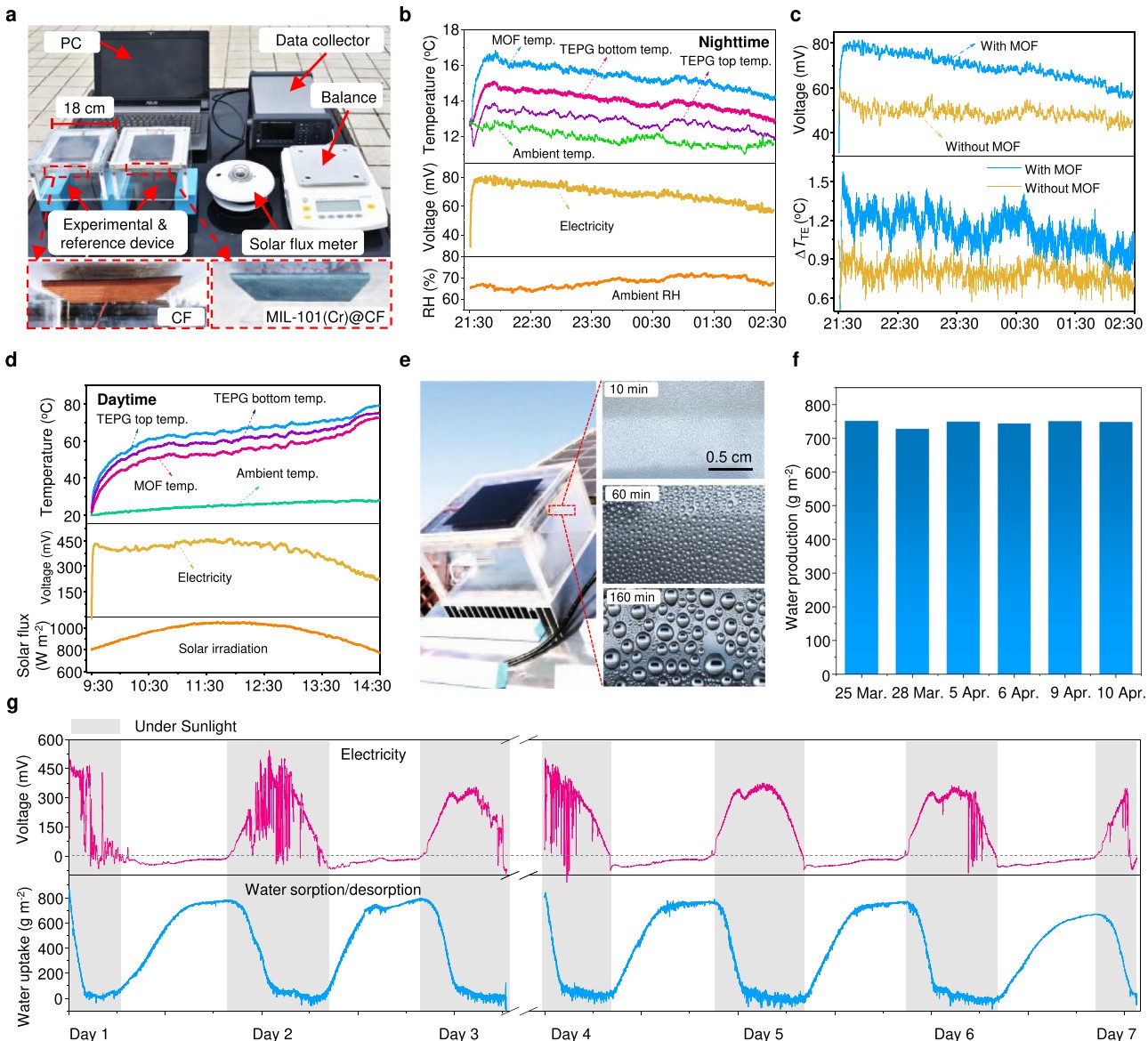

**Fig. 5 | Outdoor demonstration of the hybrid SAWH-TEPG device for water production and 24-hour power generation. a** Photograph of the hybrid SAWH-TEPG device with MIL-101(Cr)@CF (right) and the referenced device without MIL-101(Cr) (left) for outdoor experiments. **b** Real-time evolutions of air RH (bottom), open-circuit voltage (middle), ambient temperature, MOF temperature, and TEPG temperature (top) during nighttime. **c** Real-time evolutions of open-circuit voltage and temperature difference $\Delta T_{TE}$ for power generation in the hybrid device and the referenced device. **d** Real-time evolutions of solar irradiation (bottom), open-circuit voltage (middle), ambient temperature, MOF temperature, and TEPG temperature (top) during daytime. **e** Photograph of water condensation during the water collection stage. **f** Water production during repeated outdoor experiments in 2021, revealing the stable water harvesting performance of the hybrid SAWH-TEPG device. **g** Open-circuit voltage and water uptake during a continuous week of outdoor experiments.

in comparison with other recently reported devices (Supplementary Table 1). The amount of electricity and water generated by the hybrid SAWH-TEPG device can be easily scaled up and maintain stable performance by assembling numerous SAWH and TEPG units in series or parallel. The proposed moisture-induced synergistic effects, for the first time to our knowledge, not only facilitate the thermoelectric power density and atmospheric water harvesting but also realize 24-hour power generation without energy storage compared with conventional hybrid water and power co-generation systems. The power generated not only can light a LED bulb with the assistance of a voltage boost converter but also can be used to power small sensors on off-grid occasions, passively and continuously. Moreover, considering the ubiquitous moisture and natural sunlight, the hybrid SAWH-TEPG device is capable of realizing cost-effective co-generation of water and power at any time and anywhere. Our work provides

a promising approach to solving water and electricity shortages for individuals living in arid and remote regions.

## Methods

### Preparation of MIL-101(Cr)@CF composite

Chromic chloride hexahydrate ($CrCl_3 \cdot 6H_2O$), terephthalic acid ($C_6H_4$–1,4-$(CO_2H)_2$), N,N-Dimethyl formamide (DMF), ethanol, and all raw Chemicals were purchased from Sigma-Aldrich. Firstly, MIL-101(Cr) was synthesized using a hydrothermal reaction. 1 mmol chromic chloride hexahydrate and 1 mmol terephthalic acid were dissolved into 7.2 mL deionized water. The mixed solution was treated by hydrothermal reaction at 190 °C for 24 h to synthesize MIL-101(Cr). The MOF powder was obtained by removing recrystallized terephthalic acid and washing with DMF and ethanol respectively. Secondly, MIL-101(Cr)@CF was prepared using the solution impregnation

method. A porous copper foam (CF) was firstly treated by $0.5\,mol\,L^{-1}$ HCl solution and then cleaned by ethanol and deionized water with ultra-sonic cleaning. Afterwards, MIL-101(Cr) suspension was prepared by mixing 1 g MIL-101(Cr) powder with 8 mL ethanol and 2 mL water, and then 0.37 g of silicate sol (~30 wt%) was added into the suspension as binder. The as-synthesized MOF suspension was filled into the open-cell of CF and then dried at 80 °C for 24 h to form robust MOF coating on the skeleton of porous CF. The mass fraction of MIL-101(Cr) is up to 90wt % in the MIL-101(Cr)@CF composite.

## Characterizations of MIL-101(Cr)@CF composite

The morphology of MIL-101(Cr)@CF was observed by Scanning Electron Microscopy (SEM, Sirion 200 instrument, FEI) equipped with an energy-dispersive X-ray spectrometer (EDS, INCA X-Act attachment, Oxford). The nitrogen adsorption was measured by physisorption apparatus (Autosorb-IQ3, Quantachrome) and the specific surface area is determined by Brunauer-Emmett-Teller (BET) method. The thermal conductivity of MIL-101(Cr)@CF was measured by using laser flash method (LFA 447, Netzsch) and Hot Disk thermal constants analyzer (TPS3500, Hot Disk AB Company, Sweden). The PXRD patterns were measured by an X-ray diffractometer (Ultima IV, Rigaku). FT-IR spectra were measured by an FT-IR spectrometer (Nicolet 6700, Thermo Fisher Scientific). The water sorption isotherms were measured using ASAP analyzer (ASAP2020, Micromeritics) under controllable water vapor pressure. The sample was set at a constant temperature (15, 25, and 35 °C), and the relative pressure of water vapor was increased from 0 to 1 according to pressure intervals. The TGA tests were carried out by thermogravimetric analyzer (STA 449, Netzsch) equipped with a moisture humidity generator (MHG 32, ProUmid).

## Water uptake of MIL-101(Cr)@CF composite

The water uptake of MIL-101(Cr)@CF was measured using a self-constructed sorption testing device with a precision electronic balance (ME503TE, METLER TOLEDO). The MOF and its composite samples were dried at 100 °C for 4 h to determine the dry MOF mass without water content. The samples were then cooled down and placed on the electronic balance at a predetermined environment condition. The mass change of sample during water vapor sorption process was recorded continuously until reaching its sorption equilibrium. The water uptake of MIL-101(Cr)@CF ($g_{water}\,g_{sorbent}^{-1}$) is calculated by the mass difference of the composite before and after water sorption. The moisture sorption-desorption isotherms were obtained by measuring the water uptake at different relative pressures ($P\,P_0^{-1}$) for a given temperature of 25 °C. The moisture sorption/desorption heat was measured by thermogravimetric analysis coupled with differential scanning calorimetry (TGA-DSC).

## Performance of hybrid SAWH-TEPG device

Indoor and outdoor experiments were carried out to evaluate the performance of hybrid SAWH-TEPG device. The amount of water harvesting of the SAWH module was measured in real-time by a precision electronic balance (ME503TE, METLER TOLEDO) with a resolution of 1 mg, and the output voltage of the TEPG module was measured in real-time by a voltmeter. Ambient humidity and temperature were also recorded by Agilent data collector. For the indoor experiments, all proof-of-concept experiments of the hybrid device were carried out in an environment-controlled chamber and the swings of temperature and RH were kept below 0.5 °C and 2%, measured by the thermo/hygrometer (HF335, Rotronic). A water-cooling plate was used to simulate radiative cooling for accelerating the moisture sorption of the hybrid device at 25 °C and 65% RH. The moisture desorption was carried out under light intensities of 500, 750, and $1000\,W\,m^{-2}$ using a solar simulator (UHE-NS-100, SCIENCETECH) with a uniformity of 10%, temporal instability of 10%, and spectral match classification of Class AAA. For the outdoor experiments, the hybrid device was placed on

the roof of our laboratory in Shanghai, and a transparent polyethylene film was used to suppress the convective heat loss of the hybrid device. During nighttime, moisture sorption occurs from 9:30 pm to 2:30 am (UTC + 8), and radiative cooling enhances moisture sorption and power generation. During daytime, sola drives the hybrids device to perform moisture desorption from 9:30 am to 2:30 pm (UTC + 8). The solar flux was measured in real time by a pyranometer (TBQ-DL, Jinzhou Sunshine Technology). As a contrast, a referenced device without MOF sorbent was fabricated and tested to evaluate the moisture sorption/desorption effect on the performance of power generation. The temperature evolutions of the dual-functional coating layer, TEPG module, SAWH module, air temperature, and RH were measured in real-time by installing thermo/hygrometer inside the hybrid device (HF335, Rotronic).

## Data availability

All the data needed to evaluate the conclusions in the paper are present in the paper and/or the Supplementary information. Source data are provided with this paper.

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

## Acknowledgements

This work was supported by the National Natural Science Foundation of China under contract no. 51876117 and the National Key R&D Program of China under contract no. 2018YFE0100300.

## Author contributions

Conceptualization: T.L.; methodology: M.W., J.X., T.L.; investigation: M.W., R.D., T.Y., S.W.; visualization: M.W., P.W.; supervision: T.L.; Writing —original draft: T.L., M.W., J.X.; writing—review & editing: T.L., J.X., Z.B, R.W. All the authors discussed results and commented on the manuscript.

## Competing interests

The authors declare no competing interests.
