## [Peer Review File · Nature Communications]

Simultaneous Atmospheric Water Production and 24-hour Power Generation Enabled by Moisture-induced Energy HarvestingREVIEWER COMMENTS

Reviewer #1 (Remarks to the Author):

Very good technology and an inspiring approach!

Please do have the paper reviewed by a native English speaker, there are a few grammatical errors throughout.

The water production is impressive, the electric power production, while even available at night to some degree, is very encouraging but still very small. The reviewer feels that electric capability is a bit too overemphasized.

It would be great to see data of water production and power production as a function of time of day and possibly even over a year! But this may be asking too much for this paper.

Reviewer #2 (Remarks to the Author):

In this work, the authors developed a novel atmospheric water harvesting system that combines thermoelectric power generation system. Such design smartly utilizes the enthalpy consumption of desorption to cool the TEPG module, which can enhance the temperature gradient and electricity generation performance. Additionally, the heat released by the moisture sorption process is also used to generate electricity at night, and by coupling the radiative cooling, the device shows a good electricity generation performance. The design shows some promise in the field of water-energy nexus. Here are my comments:

(1) The SAWH should have enough water to ensure the performance of the SAWH-TEPG Device. is the adsorbed water enough to maintain the good performance of the TEPG? Was there a mechanism to ensure this?

(2) The authors are suggested to explain the temperature rise in Figure 5d after 12:30. More description on the experimental conditions can be helpful.

(3) In nighttime, the moisture is absorbed by the MOF and release its latent heat. Basically, the heat can be calculated by multiplying the moisture sorption rate and latent heat. The authors are suggested to present the heat power of this process to show how heat can be utilized. Further discussion is needed to educate the general audience on the niche application of this painstakingly produced electricity.

(4) In the SAWH-TEPG Device operating at night, the coating on TEPG provide a cooling power by radiative cooling, what is the cooling power of the device? Moreover, what is the cooling power of the SAWH in the daytime. The work needs to be more quantitative in many aspects and those are just some examples.

Responses to the Referees

We greatly appreciate the referees of his/her reviews and comments. Below are the point-to-point answers to the questions and suggestions raised by the referees. In this response to the referees' comments:

1. Black color - comments from the referees
2. Blue color - responses to the comments
3. Red color - added/changed to the Manuscript and Supplementary Information

Responses to Referee #1

Very good technology and an inspiring approach!

Response: We are very grateful for your encouraging comments!

Comment 1: Please do have the paper reviewed by a native English speaker, there are a few grammatical errors throughout.

Response: Thank you very much for your suggestions!

We have checked and revised the grammatical errors throughout the manuscript, and all the corrections suggested by different reviewers have been taken into account. Furthermore, the revised manuscript has been edited by a native English speaker to improve its readability.

Comment 2: The water production is impressive, the electric power production, while even available at night to some degree, is very encouraging but still very small. The reviewer feels that electric capability is a bit too overemphasized. It would be great to see data of water production and power production as a function of time of day and possibly even over a year! But this may be asking too much for this paper.

Response: Thank you very much for your comments and helpful suggestions. We performed continuous outdoor experiments for one week and supplemented the

corresponding data of water and power production as a function of local time. Meanwhile, we calculated the water harvesting efficiency of the hybrid SAWH-TEPG device based on the experimental data and predicted the annual water production of typical cities. To show the performance superiority of our hybrid SAWH-TEPG device more intuitively, we compared our power and water production results with other reported water and electricity co-generation devices from the literatures.

Fig. R1 shows the ambient conditions, as well as the corresponding open-circuit voltage and mass change of the hybrid SAWH-TEPG device during the continuous outdoor experiments. A continuous voltage output can be observed during the one-week experiments. The output voltage is proportional to the solar irradiation intensity, and the maximum voltage of 505 mV was recorded on July 31, 2022 under the solar irradiation intensity of $1029 \text{ W}\cdot\text{m}^{-2}$. As the sun goes down, the voltage output falls below zero, indicating that the MOF sorbent begins to absorb moisture from air, and the reverse open-circuit voltage is generated. Fig. R1(b) also shows the mass change of moisture adsorbed and desorbed by the MOF sorbent in the SAWH-TEPG device over one week. Similar to the voltage output, the MOF sorbent desorbs water vapor during sunlight heating and reabsorbs moisture from air after the sun goes down. The average mass change of moisture is about $800 \text{ g}\cdot\text{cm}^{-2}$, indicating that the MOF sorbent can completely adsorb or desorb water vapor during the continuous cycles in outdoor experiments.

Fig. R1. (a) Variations of ambient temperature, RH, and solar irradiation intensity during the continuous outdoor experiments. (b) Open-circuit voltage and water mass change of hybrid SAWH-TEPG device during the continuous outdoor experiments.

The real-time mass and environmental changes during the water vapor desorption and sorption processes are further illustrated in Fig. R2. The MOF sorbent begins to desorb water vapor at 9:30 a.m. and it takes about 3.5 hours to complete desorption. At 21:30, the MOF sorbent starts to absorb water vapor from air and the adsorption process lasts for approximately 7 hours.

Fig. R2. (a) Mass, solar flux, ambient temperature, and ambient RH changes of the SAWH-TEPG device during the desorption process. (b) Mass, ambient temperature, and ambient RH changes of the SAWH-TEPG device during the sorption process.

Furthermore, we evaluate solar-driven water harvesting efficiency (η) by using the following equations, and the η is determined as 21.7% based on the input solar energy and the MOF mass loss (equal to the water mass):

$$\eta = \frac{\Delta m_{\text{water}} \cdot \Delta H_{\text{H}_2\text{O}}}{Q_{\text{solar}}} \quad (1)$$

$$Q_{\text{solar}} = \int_{t_0}^{t_{\text{end}}} \alpha \cdot A \cdot q_{\text{solar}} dt \quad (2)$$

where Δm_{water} is the mass change of water by the MOF sorbent (~ 8.1 g), $\Delta H_{\text{H}_2\text{O}}$ is the water desorption enthalpy (~ 44 kJ·mol⁻¹ water), Q_{solar} is the input solar energy, α is the solar absorptance of solar collector (~ 0.95), A is the area of the solar absorber (0.1×0.1 m²), and q_{solar} is the instantaneous solar irradiation intensity.

We assessed the annual water production performance of the proposed SAWH-TEPG device under the climate data of six typical different cities. Fig. R3 shows the average annual solar irradiation intensity and relative humidity changes during one day.

The predicted annual water production (M_{water}) of our device can be calculated by the following equation:

$$M_{water} = 365 \cdot \eta \cdot Q_{solar} \quad (3)$$

Fig. R3. (a) Variation of annual average solar irradiation intensity in one day. (b) Variation of annual average relative humidity in one day.

Fig. R4. Predicted annual water production of the SAWH-TEPG device in different cities.

Fig. R4. shows the annual water production for different cities. Since the adsorption transition RH of MIL-101(Cr) is as low as 45 %, the water production performance of the SAWH-TEPG device mainly depends on the local solar irradiation intensity. As a

result, London has the lowest predicted annual water production of $295.7 \text{ L}\cdot\text{m}^{-2}\cdot\text{a}^{-1}$, and Brasilia has the highest predicted annual water production of $612.6 \text{ L}\cdot\text{m}^{-2}\cdot\text{a}^{-1}$.

The comparison of electricity power output between our device and the reported devices for simultaneous water and electricity generation is shown in Table R1. Our hybrid SAWH-TEPG device not only offers continuous power output but also provides the highest power density, opening the door to realize 24-hour power supply anytime and anywhere.

Table R1. Comparison of power output of simultaneous water and electricity generation solutions

Solar intensity ($\text{kW}\cdot\text{m}^{-2}$)	Electricity generation Methods	Daytime	Nighttime	Output type	Ref.
		Power density ($\text{W}\cdot\text{m}^{-2}$)	Power density ($\text{mW}\cdot\text{m}^{-2}$)		
1	Thermoelectric	0.4	-	Transient	[1]
1	Pyroelectric- piezoelectric	2.4×10^{-4}	-	Transient	[2]
1	Triboelectric	2.6×10^{-4}	-	Transient	[3]
1	Thermoelectric	0.175	-	Transient	[4]
1	Thermoelectric	0.4	-	Transient	[5]
1	Thermoelectric	0.22	-	Transient	[6]
0.9	Thermoelectric	0.22	-	Transient	[7]
1	Thermoelectric	0.072	-	Transient	[8]
1	Thermoelectric	0.5	-	Transient	[9]
-	Thermal resonant	0.203	< 1	Continuous	[10]
~0.9	Thermoelectric	0.52	6.6	Continuous	[11]
1	Thermoelectric	0.685	21	Continuous	This work

Changes add to manuscript:

Fig. R1(b). is added in the manuscript as **Figure 5(g)**;

The outdoor experiment reveals that the desorption cooling power is up to 300 W m^{-2} (Supplementary Figure 24), and the solar-driven water harvesting efficiency (η) is calculated as 21.7%. **(Page 20)**

We further predicted the annual water harvesting of the proposed SAWH-TEPG device under the climate data of six typical different cities (Supplementary Figure 25), showing the annual water production is 295.7-612.6 L m⁻² a⁻¹ under different climate conditions (Supplementary Figure 26). **(Page 20)**

Furthermore, we carried out continuous week-long outdoor experiments to demonstrate the simultaneous atmospheric water production and 24-hour power generation (Supplementary Figure 27). The hybrid SAWH-TEPG device exhibits continuous voltage output with maximum open-circuit voltage of 505 mV and stable water harvesting with average water uptake of 800 g·cm⁻². Notably, although the hybrid SAWH-TEPG device achieves almost complete sorption and desorption of water vapor, there is still a long equilibration duration between the two processes, suggesting that the atmospheric water production and the power generation performance can be further improved by optimizing sorbent and device structure. **(Page 20)**

Changes add to Supplementary Information (SI):

Fig. R2(b). is added in SI as Supplementary Figure 20(a)

Fig. R2(a). is added in SI as Supplementary Figure 24(a)

Fig. R3. is added in SI as Supplementary Figure 25

Fig. R4. is added in SI as Supplementary Figure 26

Responses to Referee #2

In this work, the authors developed a novel atmospheric water harvesting system that combines thermoelectric power generation system. Such design smartly utilizes the enthalpy consumption of desorption to cool the TEPG module, which can enhance the temperature gradient and electricity generation performance. Additionally, the heat released by the moisture sorption process is also used to generate electricity at night, and by coupling the radiative cooling, the device shows a good electricity generation performance. The design shows some promise in the field of water-energy nexus.

Response: We are very grateful for your encouraging comments and helpful suggestions!

Comment 1: The SAWH should have enough water to ensure the performance of the SAWH-TEPG device. Is the adsorbed water enough to maintain the good performance of the TEPG? Was there a mechanism to ensure this?

Response: Thank you very much for your comments. We understand the reviewer's concerns regarding the impact of water sorption amount of the SAWH on the performance of TEPG device. The issues raised by the reviewer can be solved by optimizing the device structure, switching to a different kinds of sorbent, or simply increasing the amount of sorbents in the device.

In this work, we put more emphasis on the concept of moisture-induced energy harvesting strategy for realizing simultaneous sorption-based atmospheric water harvesting and power generation by subtly utilizing the synergistic thermal effects of moisture sorption/desorption, radiative heating from sunlight during the day, and radiative cooling from universe during the night. A facile device was successfully developed to verify the feasibility of our proposed concept.

To ensure the good performance of TEPG under the synergistic effect of sorption heating and radiative cooling, the sorption duration of water vapor during the nighttime should assure the sorption heating for TEPG, while the desorption duration of water

vapor during the daytime should assure the desorption cooling for TEPG. As shown in Fig. R5, the SAWH device can achieve complete water vapor sorption and desorption, but there is still a long equilibrium time between these two processes, suggesting that the water harvesting and power generation performance of SAWH-TEPG device can be further enhanced.

Fig. R5. Open-circuit voltage and water mass change of hybrid SAWH-TEPG device during the continuous outdoor experiment.

Fig. R6 further shows the specific mass change of SAWH device during the desorption and sorption processes. As can be seen, the desorption and sorption durations are about 3.5 and 7 hours, respectively, indicating the desorption rate is much higher than sorption rate. It is noted that the sorption/desorption rates are strongly influenced by air RH, solar radiation, ambient temperature, thermal design of device, etc. Moreover, the above-mentioned parameters and sunlight duration always vary along with time during one day and one year.

To maintain the good performance of the TEPG, the overall guideline is to assure the sorption heating for TEPG during the whole nighttime and desorption cooling for TEPG during the whole daytime by optimizing the device structure and sorbent materials under different climate conditions. The amount of captured water from air during the nighttime is enough and can be used to assure the sufficient desorption cooling for TEPG during the daytime.

Fig. R6. (a) Mass, solar flux, ambient temperature, and ambient RH changes of the SAWH-TEPG device during the desorption process. (b) Mass, ambient temperature, and ambient RH changes of the SAWH-TEPG device during the sorption process.

Changes add to manuscript:

Fig. R5. is added in manuscript as Figure 5(g).

Furthermore, we carried out continuous week-long outdoor experiments to demonstrate the simultaneous atmospheric water production and 24-hour power generation (Supplementary Figure 27). The hybrid SAWH-TEPG device exhibits continuous voltage output with maximum open-circuit voltage of 505 mV and stable water harvesting with average water uptake of 800 g m⁻². Notably, although the hybrid SAWH-TEPG device achieves almost complete sorption and desorption of water vapor, there is still a long equilibration duration between the two processes, suggesting that the atmospheric water production and the power generation performance can be further improved by optimizing sorbent and device structure. **(Page 20)**

Changes add to Supplementary Information (SI):

Fig. R6(b). is added in SI as Supplementary Figure 20(a)

Fig. R6(a). is added in SI as Supplementary Figure 24(a)

Comment 2: The authors are suggested to explain the temperature rise in Figure 5d after 12:30. More description on the experimental conditions can be helpful.

Response: Thank you very much for your careful review and helpful suggestions. We quantitatively analyze phenomenon by developing an energy balance model for the SAWH-TEPG hybrid device, as shown in Figure R8. The energy balance model for the SAWH-TEPG device during the daytime can be expressed as follows:

Daytime:

$$P_{des} = P_{solar} - P_{rad}(T_{abs}) + P_{atm}(T_{amb}) - P_{conv}^{abs}(T_{abs}, T_{amb}) - P_{conv}^{sor}(T_{abs}, T_{amb}) \quad (4)$$

$$P_{solar} = \int d\lambda \varepsilon(\lambda, \theta_{solar}) I_{AIM1.5}(\lambda) \quad (5)$$

$$P_{rad}(T_{abs}) = \int d\Omega \cos\theta \int_0^\infty d\lambda I_{BB}(T_{abs}, \lambda) \varepsilon(\lambda, \theta) \quad (6)$$

$$P_{atm}(T_{amb}) = \int d\Omega \cos\theta \int_0^\infty d\lambda I_{BB}(T_{amb}, \lambda) \varepsilon(\lambda, \theta) \varepsilon_{atm}(\lambda, \theta) \quad (7)$$

$$P_{conv}^{abs}(T_{abs}, T_{amb}) = h_{abs}(T_{abs} - T_{amb}) \quad (8)$$

$$P_{conv}^{sor}(T_{abs}, T_{amb}) = h_{sor}(T_{sor} - T_{amb}) \quad (9)$$

where P_{des} is the desorption cooling power, P_{solar} is the incident solar irradiation absorbed by the absorber, $P_{rad}(T_{abs})$ is the power radiated by the absorber, $P_{atm}(T_{amb})$ is the absorbed atmospheric thermal radiation, $P_{conv}^{abs}(T_{abs}, T_{amb})$ is the convective power lost by the absorber, $P_{conv}^{sor}(T_{abs}, T_{amb})$ is the convective power lost by the sorbent, P_{cool}^{net} is the net radiative cooling power of the absorber.

Fig. R7. (a) Energy balance and (b) theoretical thermal equilibrium analysis of the hybrid SAWH-TEPG device during daytime.

Solar absorber (emitter) can be approximated as a graybody with average emissivity, $\varepsilon_s = 0.95$, we can express the above equations as $P_{rad}(T_{abs}) = \varepsilon_s \sigma T_{abs}^4$ and $P_{atm}(T_{amb}) = \varepsilon_{atm} \varepsilon_s \sigma T_{abs}^4$. The ε_{atm} can be defined by the correlation of $\varepsilon_{atm} = 0.741 + 0.0062 \cdot T_{dew}$. T_{dew} is the dew point temperature^[13]. h_{abs} and h_{sor} are the convective heat transfer coefficients of the absorber and the sorbent, and they are assumed to be $5 \text{ W/m}^2 \cdot \text{K}$ according to the literatures^[14-16].

We adopted the ambient data at 12:30 and 14:00 to calculate the corresponding heat flow. Fig. R8 shows the theoretical thermal equilibrium results of the hybrid SAWH-TEPG device at 12:30 and 14:00. At 12:30 pm, the radiative energy loss (P_{rad}), convective heat losses (P_{conv}^{sor} and P_{conv}^{abs}), and the desorption cooling power of the device are calculated as 337.5, 204, 149, and 279 $\text{W} \cdot \text{m}^2$, respectively. While at 14:00, radiative energy loss (P_{rad}), convective heat losses (P_{conv}^{sor} and P_{conv}^{abs}), and desorption cooling power of the device are calculated as 361.3, 223, 189.2, and 48.1 $\text{W} \cdot \text{m}^2$, respectively. The solar irradiation intensity P_{solar} for heating TEPG decreases from 969.5 to 822.1 $\text{W} \cdot \text{m}^2$ during the time from 12:00 to 14:00, however, the desorption power for cooling TEPG decreases from 279 to 48.1 $\text{W} \cdot \text{m}^2$ due to the decrease in water desorption. It can be seen that the decrease in the magnitude of cooling power is lower than that of heating power. As a result, causes the temperature rise of the device rather than temperature decrease.

Fig. R8. (a) Theoretical thermal equilibrium results of the hybrid SAWH-TEPG device at (a) at 12:30 pm and (b) at 14:00 pm.

We further performed water desorption experiments to confirm the above theoretical results. The mass change of water was real-time measured during the experiment, and the corresponding desorption cooling power was determined by the following equation:

$$P_{des} = \frac{\dot{m}_{water} \cdot \Delta H_{H_2O}}{M_{H_2O}} \quad (10)$$

where \dot{m}_{water} is the mass change rate of water desorbed by the MOF sorbent, ΔH_{H_2O} is the water desorption enthalpy ($44 \text{ kJ} \cdot \text{mol}^{-1}$ water), M_{H_2O} is the relative molecular mass of the water.

As shown in Fig. R9, the MOF sorbent began to desorb water vapor at 9:30 a.m. and completed water desorption within 3.5 hours. The maximum desorption cooling power is around $300 \text{ W} \cdot \text{m}^{-2}$ at the initial stage, and then it gradually decreases with desorption time and will become $0 \text{ W} \cdot \text{m}^{-2}$ after completing water desorption process. Interestingly, the experimental results show that desorption cooling power decreases from $192 \text{ W} \cdot \text{m}^{-2}$ at 12:00 p.m. to $8.3 \text{ W} \cdot \text{m}^{-2}$ at 14:00, which is consistent with our theoretical calculation. These results confirm that the temperature rise of the device at the end stage is mainly caused by the decrease in desorption cooling power. In fact, the temperature rise of the device was also observed in our subsequent supplementary experiments (Fig. R10).

Fig. R9. (a) Mass, solar flux, ambient temperature, and ambient RH changes of the SAWH-TEPG device during the desorption process. (b) Mass, ambient temperature, and ambient RH changes of the SAWH-TEPG device during the sorption process.

Fig. R10. Real-time evolutions of solar irradiation (bottom), ambient temperature, MOF temperature, and TEPG temperature (top) during daytime.

Changes add to manuscript:

The outdoor experiment reveals that the desorption cooling power is up to 300 W m^{-2} (Supplementary Figure 24), and the solar-driven water harvesting efficiency (η) is calculated as 21.7%. **(Page 20)**

Changes add to Supplementary Information (SI):

Fig. R7. is added in SI as Supplementary Figure 1

Fig. R8. is added in SI as Supplementary Figure 22

Fig. R9. is added in SI as Supplementary Figure 24

Supplementary Note 2.

Calculation of radiative cooling power, sorption heating power and desorption cooling power

A thermal model is developed to analyze the performance of the SAWH-TEPG device (Supplementary Figure 1). The energy balance model of the SAWH-TEPG device

during daytime and nighttime can be expressed as:

Daytime:

$$P_{des} = P_{solar} - P_{rad}(T_{abs}) + P_{atm}(T_{amb}) - P_{conv}^{abs}(T_{abs}, T_{amb}) - P_{conv}^{sor}(T_{abs}, T_{amb}) = \frac{\dot{m}_{MOF} \cdot \Delta H_{H_2O}}{M_{H_2O}} \quad (S10)$$

Nighttime:

$$P_{sor} = P_{rad}(T_{abs}) - P_{atm}(T_{amb}) + P_{conv}^{abs}(T_{abs}, T_{amb}) + P_{conv}^{sor}(T_{abs}, T_{amb}) = \frac{\dot{m}_{MOF} \cdot \Delta H_{H_2O}}{M_{H_2O}} \quad (S11)$$

$$P_{cool}^{net} = P_{rad} - P_{atm}(T_{amb}) \quad (S12)$$

where P_{des} is the desorption cooling power, P_{solar} is the incident solar irradiation absorbed by the absorber, P_{sor} is the sorption heating power, $P_{rad}(T_{abs})$ is the power radiated by the absorber, $P_{atm}(T_{amb})$ is the absorbed atmospheric thermal radiation, $P_{conv}^{abs}(T_{abs}, T_{amb})$ is the convective power lost by the absorber, $P_{conv}^{sor}(T_{abs}, T_{amb})$ is the convective power lost by the sorbent, P_{cool}^{net} is the net radiative cooling power of the absorber. \dot{m}_{MOF} is the mass change rate (water sorption or desorption rate) of the MOF sorbent, ΔH_{H_2O} is the water sorption enthalpy ($44 \text{ kJ} \cdot \text{mol}^{-1}$ water), M_{H_2O} is the relative molecular mass of water.

In the energy balance models, we approximate that the total heat flows are dominated by sorption heat and neglect heat generation and absorption due to the Seebeck effect and Joule heating in the thermoelectric legs. This simplifying approximation is made due to the power conversion efficiency of the thermoelectric module is expected to be well below 0.5%. Therefore, the above parameters can be calculated by the following equations:

$$P_{solar} = \int d\lambda \varepsilon(\lambda, \theta_{solar}) I_{AIM1.5}(\lambda) \quad (S13)$$

$$P_{rad}(T_{abs}) = \int d\Omega \cos\theta \int_0^\infty d\lambda I_{BB}(T_{abs}, \lambda) \varepsilon(\lambda, \theta) \quad (S14)$$

$$P_{atm}(T_{amb}) = \int d\Omega \cos\theta \int_0^\infty d\lambda I_{BB}(T_{amb}, \lambda) \varepsilon(\lambda, \theta) \varepsilon_{atm}(\lambda, \theta) \quad (S15)$$

$$P_{conv}^{abs}(T_{abs}, T_{amb}) = h_{abs}(T_{abs} - T_{amb}) \quad (S16)$$

$$P_{conv}^{sor}(T_{abs}, T_{amb}) = h_{sor}(T_{sor} - T_{amb}) \quad (S17)$$

where $\int d\Omega = 2\pi \int_0^{2\pi} d\theta \sin\theta$ is the angular integral over a hemisphere. $I_{BB}(T, \lambda) = \frac{2hc^2}{\lambda^5} \frac{1}{e^{hc/(\lambda k_B T)} - 1}$ is the spectral radiance of a blackbody at temperature T , h is Planck's constant, k_B is the Boltzmann constant, and c is the light speed. $\varepsilon(\lambda, \theta)$ is the directional emissivity of the surface at wavelength λ . For simplicity, solar absorber (emitter) can be approximated as a graybody with average emissivity $\varepsilon_s = 0.95$ we can express the above equations as $P_{rad}(T_{abs}) = \varepsilon_s \sigma T_{abs}^4$ and $P_{atm}(T_{amb}) = \varepsilon_{atm} \varepsilon_s \sigma T_{abs}^4$. The ε_{atm} can be defined by the correlation of $\varepsilon_{atm} = 0.741 + 0.0062 \cdot T_{dew}$. T_{dew} is the dew point temperature. h_{abs} and h_{sor} are the convective heat transfer coefficients of the absorber and the sorbent, and they are assumed to be $5 \text{ W} \cdot \text{m}^{-2} \cdot \text{K}^{-1}$ according to the literatures [1-3].

Comment 3: In nighttime, the moisture is absorbed by the MOF and release its latent heat. Basically, the heat can be calculated by multiplying the moisture sorption rate and latent heat. The authors are suggested to present the heat power of this process to show how heat can be utilized. Further discussion is needed to educate the general audience on the niche application of this painstakingly produced electricity.

Response: We greatly appreciate your comments and suggestions. We present the sorption heating power according to your helpful suggestions. Based on the mass change rate of the MOF sorbent (equal to the mass change of water), the sorption heating power (P_{sor}) of the MOF sorbent is calculated by the following equation:

$$P_{sor} = \frac{\dot{m}_{\text{water}} \cdot \Delta H_{\text{H}_2\text{O}}}{M_{\text{H}_2\text{O}}} \quad (11)$$

where \dot{m}_{water} is the mass change rate of water adsorbed by the MOF sorbent, $\Delta H_{\text{H}_2\text{O}}$ is the water sorption enthalpy ($44 \text{ kJ}\cdot\text{mol}^{-1}$), $M_{\text{H}_2\text{O}}$ is the relative molecular mass of water.

Fig. R11 shows the mass change of water sorption by the MOF sorbent and its calculated sorption heating power during the sorption process. The maximum sorption heating power is around $137 \text{ W}\cdot\text{m}^{-2}$ at initial stage, and the heating power gradually decreases along with sorption time. It will eventually drop to $0 \text{ W}\cdot\text{m}^{-2}$ when the sorbent reaches its equilibrium saturation, where sorption will stop.

Fig. R11. (a) Mass, ambient temperature, and ambient RH changes of the device during the sorption process. (b) Sorption heating power of the sorbent during the sorption process.

In addition, we also performed water desorption experiment to reveal the desorption cooling power of the SAWH during the daytime. In the experiment, the mass change of the SAWH device was real-time measured, and the corresponding desorption cooling power was calculated by the following equation:

$$P_{des} = \frac{\dot{m}_{\text{water}} \cdot \Delta H_{\text{H}_2\text{O}}}{M_{\text{H}_2\text{O}}} \quad (12)$$

where \dot{m}_{water} is the mass change rate of water desorbed by the MOF sorbent, $\Delta H_{\text{H}_2\text{O}}$ is the water desorption enthalpy ($44 \text{ kJ}\cdot\text{mol}^{-1}$), $M_{\text{H}_2\text{O}}$ is the relative molecular mass of the water.

Fig. R12. shows the mass change of water and desorption cooling power during the desorption process. The maximum desorption cooling power is about $300 \text{ W}\cdot\text{m}^{-2}$ at the initial stage, and then it gradually decreases with desorption time, and further decreases to $0 \text{ W}\cdot\text{m}^{-2}$ after completing water desorption.

Fig. R12. (a) Mass, solar flux, ambient temperature, and ambient RH changes of the SAWH-TEPG device during the desorption process. (b) Desorption cooling power of the MOF sorbent during the desorption process.

It is well known that thermoelectric devices suffer from low output power and poor energy conversion efficiency. Compared with other reported simultaneous water and electricity generation devices, our hybrid SAWH-TEPG device not only provides continuous power output, but also offers the higher power density ranging from $10^{-2} \sim 1 \text{ W} \cdot \text{m}^{-2}$ (Table R2), providing a potential route to realize 24-hour power supply anytime and anywhere. Thus, the electricity generated by the hybrid SAWH-TEPG device is sufficient for powering low-power-consumption^[12]. Besides, associating numerous devices in series or parallel, or coupling the devices with an energy storage device such as a supercapacitor, can effortlessly scale up the output power and maintain a stable power output, which may further broaden the application of TEPG in LCD, LED, Internet-of-things (IoTs), sensors, devices collecting big data, etc.

Table R2. Comparison of power output of simultaneous water and electricity generation devices

Solar intensity ($\text{kW} \cdot \text{m}^{-2}$)	Daytime		Nighttime	Output type	Ref.
	Electricity generation Methods	Power density ($\text{W} \cdot \text{m}^{-2}$)	Power density ($\text{mW} \cdot \text{m}^{-2}$)		
1	Thermoelectric	0.4	-	Transient	[1]
1	Pyroelectric-piezoelectric	2.4×10^{-4}	-	Transient	[2]
1	Triboelectric	2.6×10^{-4}	-	Transient	[3]
1	Thermoelectric	0.175	-	Transient	[4]
1	Thermoelectric	0.4	-	Transient	[5]

1	Thermoelectric	0.22	-	Transient	[6]
0.9	Thermoelectric	0.22	-	Transient	[7]
1	Thermoelectric	0.072	-	Transient	[8]
1	Thermoelectric	0.5	-	Transient	[9]
-	Thermal resonant	0.203	< 1	Continuous	[10]
~0.9	Thermoelectric	0.52	6.6	Continuous	[11]
1	Thermoelectric	0.685	21	Continuous	This work

Changes add to manuscript:

The radiative cooling power is up to 90 W m^{-2} , and sorption heating power by the MOF is up to 130 W m^{-2} (Supplementary Figure 20). **(Page 19)**

The amount of electricity and water generated by the hybrid SAWH-TEPG device can be easily scaled up and maintain stable performance by assembling numerous SAWH and TEPG units in series or parallel. **(Page 21)**

Changes add to Supplementary Information (SI):

Fig. R11. is added in SI as Supplementary Figure 20

Fig. R12. is added in SI as Supplementary Figure 24

Comment 4: In the SAWH-TEPG device operating at night, the coating on TEPG provide a cooling power by radiative cooling, what is the cooling power of the device? Moreover, what is the cooling power of the SAWH in the daytime. The work needs to be more quantitative in many aspects and those are just some examples.

Response: Thank you very much for your comments and helpful suggestions.

As shown in Fig. R13, a thermal model is developed to analyze the performance of the device in the experiment. The energy balance model of the SAWH-TEPG device during daytime and nighttime can be expressed as:

Daytime:

$$P_{des} = P_{solar} - P_{rad}(T_{abs}) + P_{atm}(T_{amb}) - P_{conv}^{abs}(T_{abs}, T_{amb}) - P_{conv}^{sor}(T_{abs}, T_{amb}) = \frac{\dot{m}_{MOF} \cdot \Delta H_{H_2O}}{M_{H_2O}} \quad (13)$$

Nighttime:

$$P_{sor} = P_{rad}(T_{abs}) - P_{atm}(T_{amb}) + P_{conv}^{abs}(T_{abs}, T_{amb}) + P_{conv}^{sor}(T_{abs}, T_{amb}) = \frac{\dot{m}_{MOF} \cdot \Delta H_{H_2O}}{M_{H_2O}} \quad (14)$$

$$P_{cool}^{net} = P_{rad} - P_{atm}(T_{amb}) \quad (15)$$

where P_{des} is the desorption cooling power, P_{solar} is the incident solar irradiation absorbed by the absorber, P_{sor} is the sorption heating power, $P_{rad}(T_{abs})$ is the power radiated by the absorber, $P_{atm}(T_{amb})$ is the absorbed atmospheric thermal radiation, $P_{conv}^{abs}(T_{abs}, T_{amb})$ is the convective power lost by the absorber, $P_{conv}^{sor}(T_{abs}, T_{amb})$ is the convective power lost by the sorbent, P_{cool}^{net} is the net radiative cooling power of the absorber. \dot{m}_{MOF} is the mass change rate (water sorption or desorption rate) of the MOF sorbent, ΔH_{H_2O} is the water sorption enthalpy ($44 \text{ kJ} \cdot \text{mol}^{-1}$), M_{H_2O} is the relative molecular mass of water.

Fig. R13. (a) Energy balance and (b) theoretical thermal equilibrium analysis of the hybrid SAWH-TEPG device during daytime. (c) Energy balance and (d) theoretical thermal equilibrium analysis of the hybrid SAWH-TEPG device during nighttime.

In the energy balance models, we approximate that the total heat flows are dominated by sorption heat and neglect heat generation and absorption due to the Seebeck effect and Joule heating in the thermoelectric legs. This simplifying approximation is made due to the power conversion efficiency of the thermoelectric module is expected to be well below 0.5%. Therefore, the above parameters can be calculated by the following equations:

$$P_{solar} = \int d\lambda \varepsilon(\lambda, \theta_{solar}) I_{AIM1.5}(\lambda) \quad (16)$$

$$P_{rad}(T_{abs}) = \int d\Omega \cos\theta \int_0^\infty d\lambda I_{BB}(T_{abs}, \lambda) \varepsilon(\lambda, \theta) \quad (17)$$

$$P_{atm}(T_{amb}) = \int d\Omega \cos\theta \int_0^\infty d\lambda I_{BB}(T_{amb}, \lambda) \varepsilon(\lambda, \theta) \varepsilon_{atm}(\lambda, \theta) \quad (18)$$

$$P_{conv}^{abs}(T_{abs}, T_{amb}) = h_{abs}(T_{abs} - T_{amb}) \quad (19)$$

$$P_{conv}^{sor}(T_{abs}, T_{amb}) = h_{sor}(T_{sor} - T_{amb}) \quad (20)$$

where $\int d\Omega = 2\pi \int_0^{2\pi} d\theta \sin\theta$ is the angular integral over a hemisphere. $I_{BB}(T, \lambda) = \frac{2hc^2}{\lambda^5} \frac{1}{e^{hc/(\lambda\kappa_B T)} - 1}$ is the spectral radiance of a blackbody at temperature T , h is Planck's constant, κ_B is the Boltzmann constant, and c is the speed of light. $\varepsilon(\lambda, \theta)$ is the directional emissivity of the surface at wavelength λ . For simplicity, since our solar absorber (emitter) can be approximated as a graybody with average emissivity $\varepsilon_s = 0.95$ we can express the above equations as $P_{rad}(T_{abs}) = \varepsilon_s \sigma T_{abs}^4$ and $P_{atm}(T_{amb}) = \varepsilon_{atm} \varepsilon_s \sigma T_{abs}^4$. The ε_{atm} can be defined by the correlation of $\varepsilon_{atm} = 0.741 + 0.0062 \cdot T_{dew}$. T_{dew} is the dew point temperature^[13]. h_{abs} and h_{sor} are respectively the convective heat transfer coefficients of the absorber and the sorbent, and they are assumed to be $5 \text{ W} \cdot \text{m}^{-2} \cdot \text{K}^{-1}$ according to the literatures^[14-16].

Based on the above models, the radiative cooling power of the SAWH-TEPG device is shown in Fig. R14. The SAWH-TEPG device with MOF sorbent exhibits higher radiative cooling power compared to the TEPG device without MOF sorbent, indicating that the synergistic effect of sorption heating and radiative cooling increases the heat flow through the TEPG, and thus improves the power output performance of TEPG.

Fig. R14. Comparison of the radiative cooling power of the device with and without sorbent.

In addition, we quantify the energy flow of the SAWH-TEPG device at a different times to better illustrate the energy utilization characteristics of this device. Fig. R15 and Fig. R16 show the theoretical thermal equilibrium results of the hybrid SAWH-TEPG device during water sorption (22:30 and 01:30) and desorption processes (12:30 and 14:00) processes. During the water sorption process, the sorption heating power keeps stable (116.2 W·m⁻² to 112.5 W·m⁻²), indicating that the sorbent maintains a stable sorption rate. However, during water desorption, the desorption cooling power decrease from 279 W·m⁻² to 48.1 W·m⁻², indicating that the sorbent almost complete desorption after 1.5 h under solar-thermal heating. The above theoretical results are consistent with the experimental results obtained by measuring the mass change rates of the sorbent during the water sorption and desorption process (Fig. R17).

Fig. R15. (a) Theoretical thermal equilibrium results of the hybrid SAWH-TEPG device at (a) 22:30 and (b) 01:00 am.

Fig. R16. (a) Theoretical thermal equilibrium results of the hybrid SAWH-TEPG device at (a) at 12:30 pm and (b) at 14:00 pm.

Fig. R17. (a) Sorption heating power of the MOF sorbent during the desorption process. (b) Desorption cooling power of the MOF sorbent during the desorption process.

Furthermore, we evaluate solar-driven water harvesting efficiency (η) by using the following equations, and the η is determined as 21.7% based on the input solar energy and the MOF mass loss (equal to the water mass):

$$\eta = \frac{\Delta m_{\text{water}} \cdot \Delta H_{\text{H}_2\text{O}}}{Q_{\text{solar}}} \quad (21)$$

$$Q_{\text{solar}} = \int_{t_0}^{t_{\text{end}}} \alpha \cdot A \cdot q_{\text{solar}} dt \quad (22)$$

where Δm_{water} is the mass change of water by the MOF sorbent (~ 8.1 g), $\Delta H_{\text{H}_2\text{O}}$ is the water desorption enthalpy (~ 44 kJ mol⁻¹ water), Q_{solar} is the input solar energy, α is the solar absorptance of solar collector (~ 0.95), A is the area of the solar absorber (0.1×0.1 m²), and q_{solar} is the instantaneous solar irradiation intensity.

We assessed the annual water production performance of the proposed SAWH-TEPG device under the climate data of six typical different cities. Fig. R18 shows the average annual solar irradiation intensity and relative humidity changes during one day. The predicted annual water production (M_{water}) of our device can be calculated by the following equation:

$$M_{water} = 365 \cdot \eta \cdot Q_{solar} \quad (23)$$

Fig. R19. Shows the predicted annual water production in different cities. Since the adsorption transition RH of MIL-101(Cr) is as low as 45 %, the water production performance of the SAWH-TEPG device mainly depends on the local solar irradiation intensity. The annual water production is 295.7-612.6 L·m⁻²·a⁻¹ under different climate conditions.

Fig. R18. (a) Variation of annual average solar irradiation intensity in one day. (b) Variation of annual average relative humidity in one day.

Fig. R19. Predicted annual water production of the SAWH-TEPG device in different cities.

Changes add to manuscript:

These results indicate the sorption heating from the MOF sorbent is higher than the radiative cooling from the universe. Moreover, we found the radiative cooling power becomes higher when compared with the reference device enabled by the sorption heating of the MOF because its higher cold-side temperature enlarges the temperature difference for radiative heat transfer between TEPG and the universe (Supplementary Figure 18). We further analyze radiative cooling power and sorption heating power during the nighttime (Supplementary Figure 19, Supplementary Note 2), showing radiative cooling power is up to 90 W m^{-2} , and sorption heating power by the MOF is up to 130 W m^{-2} (Supplementary Figure 20). **(Page 19)**

The outdoor experiment reveals that the desorption cooling power is up to 300 W m^{-2} (Supplementary Figure 24), and the solar-driven water harvesting efficiency (η) is calculated as 21.7%. **(Page 20)**

We further predicted the annual water harvesting of the proposed SAWH-TEPG device under the climate data of six typical different cities (Supplementary Figure 25), showing

the annual water production is 295.7-612.6 L m⁻² a⁻¹ under different climate conditions (Supplementary Figure 26). **(Page 20)**

Furthermore, we carried out continuous week-long outdoor experiments to demonstrate the simultaneous atmospheric water production and 24-hour power generation (Supplementary Figure 27). The hybrid SAWH-TEPG device exhibits continuous voltage output with maximum open-circuit voltage of 505 mV and stable water harvesting with average water uptake of 800 g cm⁻². Notably, although the hybrid SAWH-TEPG device achieves almost complete sorption and desorption of water vapor, there is still a long equilibration duration between the two processes, suggesting that the atmospheric water production and the power generation performance can be further improved by optimizing sorbent and device structure.

Changes add to Supplementary Information (SI):

Fig. R13. is added in SI as Supplementary Figure 1

Fig. R14. is added in SI as Supplementary Figure 18

Fig. R15. is added in SI as Supplementary Figure 19

Fig. R16. is added in SI as Supplementary Figure 22

Fig. R17(a). is added in SI as Supplementary Figure 20(b)

Fig. R17(b). is added in SI as Supplementary Figure 24(b)

Finally, we authors thank the editor and referees again for their valuable comments and constructive suggestions!

References

- [1] Zhang Y, Ravi SK, Tan SC. Food-derived carbonaceous materials for solar desalination and thermo-electric power generation. *Nano energy*. 2019;65:104006.
- [2] Zhu L, Gao M, Peh CKN, et al. Self-contained monolithic carbon sponges for solar-driven interfacial water evaporation distillation and electricity generation. *Advanced Energy Materials*. 2018;8:1702149.
- [3] Gao M, Peh CK, Phan HT, et al. Solar absorber gel: localized macro-nano heat channeling for efficient plasmonic Au nanoflowers photothermic vaporization and triboelectric generation. *Advanced Energy Materials*. 2018;8:1800711.
- [4] Jiang H, Ai L, Chen M, et al. Broadband Nickel Sulfide/Nickel Foam Based Solar Evaporator for Highly Efficient Water Purification and Electricity Generation. *ACS Sustainable Chemistry & Engineering*. 2020;8:10833-10841.
- [5] Zhu L, Ding T, Gao M, et al. Shape Conformal and Thermal Insulative Organic Solar Absorber Sponge for Photothermal Water Evaporation and Thermoelectric Power Generation. *Advanced Energy Materials*. 2019;9:1900250.1-7.
- [6] Zhang X, Gao W, Su X, et al. Conversion of solar power to chemical energy based on carbon nanoparticle modified photo-thermoelectric generator and electrochemical water splitting system. *Nano energy*. 2018:S2211285518301940.
- [7] Zhang Q, Chen S, Fu Z, et al. Temperature-difference-induced electricity during solar desalination with bilayer MXene-based monoliths. *Nano energy*. 2020;76:105060.
- [8] Li N, Yang DJ, Shao Y, et al. Nanostructured Black Aluminum Prepared by Laser Direct Writing as a High-Performance Plasmonic Absorber for Photothermal/Electric Conversion. *ACS applied materials & interfaces*. 2021;13:4305-4315.
- [9] Liu X, Mishra DD, Li Y, et al. Biomass-Derived Carbonaceous Materials with Multichannel Waterways for Solar-Driven Clean Water and Thermoelectric Power Generation. *ACS Sustainable Chemistry & Engineering*. 2021;12:4571-4582.
- [10] Cottrill AL, Liu AT, Kunai Y, et al. Ultra-high thermal effusivity materials for resonant ambient thermal energy harvesting. *Nature communications*. 2018;9:1-11.
- [11] Yang K, Pan T, Pinnau I, et al. Simultaneous generation of atmospheric water and electricity using a hygroscopic aerogel with fast sorption kinetics. *Nano energy*. 2020;78:105326.
- [12] Shen D, Duley WW, Peng P, et al. Moisture-enabled electricity generation: from physics and materials to self-powered applications. *Advanced Materials*. 2020;32:2003722.
- [13] Berdahl P, Martin M, Sakkal F. Thermal performance of radiative cooling panels. *International Journal of Heat and Mass Transfer*. 1983;26:871-80.
- [14] Zhao D, Aili A, Zhai Y, et al. Radiative sky cooling: Fundamental principles, materials, and applications. *Applied Physics Reviews*. 2019;6:021306.
- [15] Wang T, Wu Y, Shi L, et al. A structural polymer for highly efficient all-day passive radiative cooling. *Nature communications* 2021;12:356.
- [16] Raman AP, Li W, Fan S. Generating Light from Darkness - ScienceDirect. *Joule*. 2019;3:11.

REVIEWER COMMENTS

Reviewer #2 (Remarks to the Author):

I have no further comments.

Reviewer #3 (Remarks to the Author):

This paper describes an experimental work with lab and field demonstrations of a not-in-kind water harvesting and power generation system for arid locations, leveraging sorption and thermoelectric principles. This type of study is of interest to the thermal-fluids and climate communities as they strive to make advances in the art and carve paths towards a more sustainable future. The paper is well written, minor revision is recommended. The authors provided a good introductory background. The conclusions are supported by the measured data. Figures quality can be improved, some are too polluted or too small, or have low resolution. It is recommended to work on the images. It would be nice to have better pictures of the actual device. The supplemental material was very helpful. The reviewer recommends the paper to be accepted with minor revisions.

The proposed system seems scientifically sound, however some additional discussion on the following is recommended.

- How is thermodynamic efficiency defined and how it compares to the other technologies listed in the introduction?
- While generating electricity is certainly a pro, there is a trade-off between water production and power generation that needs to be addressed. Considering the potential of each, it seems as though the water production has a greater value since the power generation is in the order of mW/m^2 , which means you need $5\text{-}10\text{m}^2$ to light up an LED bulb
- The atmospheric heat source/sink is poorly described. Is the air-cooled condenser driven by forced or natural convection? How much surface area is needed for that?

Additional comments

- Figure 1: recommend labeling a, b, c... from top to bottom, so the reader can follow the daytime process first, then the nighttime's
- Figure 5: it could be split into separate figures so each can be enlarged, it is difficult to read legends and see the details
- Page 20, line 416: how is the efficiency calculated
- Supplementary figures 1/19/22 are misleading: the thermal network diagrams seem to suggest a heat flow based on temperature levels on a vertical scale, but the temperature locations don't correspond to their levels (e.g., T_{universe} should be all the way to the bottom; it looks like $T_{\text{sor}}=T_{\text{abs}}$ which means there can be no heat flow). I recommend not using this figure in the paper, alternatively, it needs to be revised

Responses to the referees

We are grateful for the time and effort that reviewers have spent in carefully reviewing our manuscript. Your comments and suggestions are very helpful for us to further improving the quality of manuscript. Below are the point-to-point answers to the questions and suggestions raised by the referees. In this response to the referees' comments:

1. Black color - comments from the referees
2. Blue color - responses to the comments
3. Red color - added/changed to the Manuscript and Supplementary Information

Reviewer #2:

I have no further comments.

Response: Thank you very much for your review and support!

Reviewer #3:

This paper describes an experimental work with lab and field demonstrations of a not-in-kind water harvesting and power generation system for arid locations, leveraging sorption and thermoelectric principles. This type of study is of interest to the thermal-fluids and climate communities as they strive to make advances in the art and carve paths towards a more sustainable future. The paper is well written, minor revision is recommended. The authors provided a good introductory background. The conclusions are supported by the measured data. Figures quality can be improved, some are too polluted or too small, or have low resolution. It is recommended to work on the images. It would be nice to have better pictures of the actual device. The supplemental material was very helpful. The reviewer recommends the paper to be accepted with minor revisions.

Response: Thank you very much for your careful review and positive comments!

Figures quality were improved according to your good suggestions. Below are the detailed point-to-point response to your comments.

The proposed system seems scientifically sound, however some additional discussion on the following is recommended.

1. How is thermodynamic efficiency defined and how it compares to the other technologies listed in the introduction?

Response: Thank you very much for your comments!

The thermodynamic efficiency of atmospheric water harvesting technology generally refers to the thermal efficiency (η_{thermal}), which was calculated as,

$$\eta_{\text{thermal}} = \frac{m_{\text{water}} \Delta H_{\text{water}}}{Q_{\text{in}}}$$

where the m_{water} represents the mass of harvested water, the ΔH_{water} is the enthalpy (latent heat) of water condensation, and the Q_{in} is the amount of input energy. For solar-driven atmospheric water harvesting, the Q_{in} means the input solar irradiation.

To quantify the intrinsic capability of water productivity in view of energy consumption, the specific water yielding per unit of input energy (SY_{water}), also can be expressed as the specific energy consumption (SEC) per unit of produced water, is also commonly used to evaluate the energy

efficiency of water production devices, which is scaled with η by the proportional constant ΔH_{water} and calculated as,

$$SY_{\text{water}} = \frac{m_{\text{water}}}{Q_{\text{in}}}$$

$$SEC = \frac{Q_{\text{in}}}{m_{\text{water}}}$$

where the unit of SY_{water} is $L_{\text{water}}/\text{kWh}$ and the unit of SEC is $\text{kWh}/L_{\text{water}}$, respectively.

Either η_{thermal} , SY_{water} , or SEC can be used to compare the energy efficiency of different water production technologies. For desalination driven by thermal energy such as multi-stage flash and distillation, its SY_{water} is about 42-83 $L_{\text{water}}/\text{kWh}$ (Desalination, 495, 114659, 2020), whose η_{thermal} can be higher than 100% by the heat recovery (Energy Environ. Sci., 14, 1771–1793, 2021). For the water harvesting from air based on the direct air cooling and condensation, its SY_{water} is about 1.1-4.2 $L_{\text{water}}/\text{kWh}$ at RH above 50%RH, while that of the sorption-based AWH is about 0.12-0.75 $L_{\text{water}}/\text{kWh}$ (Environmental Science: Water Research & Technology, 6, 2016-2034, 2020).

As we discussed in the Introduction section, the efficiency of current AWH is relatively lower than other technologies; thus, it is of great importance to develop more efficient AWH systems by advanced thermal design and energy utilization strategy. Therefore, many researchers focus on improving thermal efficiency, such as research group of Evelyn N. Wang from MIT proposed a dual-stage energy utilization strategy and achieved a relatively high thermal efficiency of 9% (Joule, 4, 1-17, 2020). Although the AWH technology has relatively low efficiency and water productivity, its inherent special advantage of being accessible anytime and anywhere indicates its potential application in remote and arid regions; thus, has great research values.

Changes add to manuscript:

The poor performance not only can be ascribed to the low water sorption capacity of sorbents at low RH, but also as a result of the low thermal efficiency of solar-driven SAWH devices. (Page 3)

The solar-driven water harvesting efficiency (η) is calculated as 21.7% by the following equation,

$$\eta = \frac{m_{\text{water}} \Delta H_{\text{water}}}{\int_{t_0}^{t_{\text{end}}} W_{\text{solar}} dt}$$

where the m_{water} represents the mass of harvested water, the ΔH_{water} is the enthalpy (latent heat) of water condensation, and the W_{solar} is the real-time solar irradiation intensity during the water production process. (Page 20)

2. While generating electricity is certainly a pro, there is a trade-off between water production and power generation that needs to be addressed. Considering the potential of each, it seems as though the water production has a greater value since the power generation is in the order of mW/m^2 , which means you need 5-10 m^2 to light up an LED bulb

Response: Thank you very much for your comment!

We agree your viewpoint that the current power density is relatively low, which is the common challenge for all thermoelectric devices driven by low temperature differences in this research field. In this work, we make a one step forward to improving the power density by coupling AWH system and achieved one of the highest power densities in comparison with other works (Table R1).

Table R1. Comparison of power output of simultaneous water and electricity generation solutions

Solar intensity (kW/m ²)	Electricity generation Methods	Daytime	Nighttime	Output type	Ref.
		Power density (W/m ²)	Power density (mW/m ²)		
1	Thermoelectric	0.4	-	Transient	[1]
1	Pyroelectric-piezoelectric	2.4×10 ⁻⁴	-	Transient	[2]
1	Triboelectric	2.6×10 ⁻⁴	-	Transient	[3]
1	Thermoelectric	0.175	-	Transient	[4]
1	Thermoelectric	0.4	-	Transient	[5]
1	Thermoelectric	0.22	-	Transient	[6]
0.9	Thermoelectric	0.22	-	Transient	[7]
1	Thermoelectric	0.072	-	Transient	[8]
1	Thermoelectric	0.5	-	Transient	[9]
-	Thermal resonant	0.203	< 1	Continuous	[10]
~0.9	Thermoelectric	0.52	6.6	Continuous	[11]
1	Thermoelectric	0.685	21	Continuous	This work

We also agree that there is a trade-off between water production and power generation, but the energy consumption by power generation is much smaller than that by water harvesting; thus, the power generation has generable influence on daytime water release. Especially, the solar collection temperature is high enough to drive both of the power generation and water release due to the low water desorption temperature of MOFs. Furthermore, the added thermoelectric modular make a positive impact on promoting the water sorption at night by extra heat absorption of power generation. To light up a LED bulb, a DC-DC voltage boost converter can be used, as reported by other researchers (Joule, 3, 2679-2686, 2019) who use a 25 mW/m² thermoelectric modular lighted up a LED (Jameco 3 mm 8,0001 mcd White LED). As your suggestion, we make more discussion on this point in the revised manuscript.

Changes add to manuscript:

More importantly, both daytime and nighttime synergistic effects enable the hybrid SAWH-TEPG device to exhibit high water production of 750 g m⁻² and impressive all-day thermoelectric generation up to 685 mW m⁻² in the daytime and 21 mW m⁻² in the nighttime, achieving one of highest power density in comparison with other recently reported devices (Table S1). (Page 21)

The power generated not only can light a LED bulb with the assistance of a voltage boost converter but also can be used to power small sensors on off-grid occasions, passively and continuously. (Page 22)

Changes add to SI file:

Table R1 and related references were added in SI file as Table S1.

3. The atmospheric heat source/sink is poorly described. Is the air-cooled condenser driven by forced or natural convection? How much surface area is needed for that?

Response: Thank you very much for your comments!

We are sorry for the insufficient description on the heat sink. In this work, we used the ambient air as heat sink to cool down the water condenser, which was driven by natural convective heat transfer for the passive AWH without extra electricity consumption. In comparison with the forced air convection, natural convection has relative low convective heat transfer coefficient of $\sim 10 \text{ W}/(\text{m}^2 \cdot \text{K})$. Therefore, to realize the timely dissipation of condensation heat, we designed an aluminum radiator with flat fins as water condenser ($17 \text{ cm} * 17 \text{ cm} * 4.5 \text{ cm}$), as shown in Fig. R1.

Fig. R1. Photograph of air-cooled condenser using aluminum flat fins.

The energy balance of air-cooled condenser during water condensation process can be expressed as following equation,

$$\frac{dm_{\text{H}_2\text{O}}}{dt} \Delta H_{\text{H}_2\text{O}} = h\eta_0 A_{\text{con}} (T_{\text{con}} - T_{\text{ambient}})$$

where the $m_{\text{H}_2\text{O}}$ is the mass of water condensed on the air-cooled condenser, the $\Delta H_{\text{H}_2\text{O}}$ is the condensation heat per gram water, the T_{con} represents the temperature of condenser, the h represents the natural heat convection coefficient estimated as $10 \text{ W}/(\text{m}^2 \cdot \text{K})$, and the η_0 represents the overall fin surface efficiency, which is considered as 1 because natural heat convection is much slower than the thermal conduction of aluminum ($272 \text{ W}/\text{m} \cdot \text{K}$). According to the water harvesting rate of $150 \text{ g}/(\text{m}^2 \cdot \text{h})$, the required surface area of condenser is 0.092 m^2 to keep a condensation temperature no more $1 \text{ }^\circ\text{C}$ higher than ambient. Therefore, we employed the aluminum radiator with fins (Figure R1) as water condenser, whose heat exchange surface area is 0.247 m^2 , provides enough heat exchange surface to efficiently dissipate the condensation heat release by water vapor.

Changes add to manuscript:

The released water vapor condensates on the surface of an air-cooled aluminum condenser with

natural convective heat transfer (Figure S25), and then becomes water droplets to be collected (Figure 5e). (Page 20)

Changes add to SI file:

Fig. R1 were added in SI file as Figure S25.

The energy balance of air-cooled condenser during water condensation process can be expressed as following equation,

$$\frac{dm_{H_2O}}{dt} \Delta H_{H_2O} = h\eta_0 A_{con} (T_{con} - T_{ambient})$$

where the m_{H_2O} is the mass of water condensed on the air-cooled condenser, the ΔH_{H_2O} is the condensation heat per gram water, the T_{con} represents the temperature of condenser, the h represents the natural heat convection coefficient estimated as $10 \text{ W}/(\text{m}^2 \cdot \text{K})$, and the η_0 represents the overall fin surface efficiency, which is considered as 1 because natural heat convection is much slower than the thermal conduction of aluminum ($272 \text{ W}/\text{m} \cdot \text{K}$). According to the water harvesting rate of $150 \text{ g}/(\text{m}^2 \text{h})$, the required surface area of condenser is 0.092 m^2 to keep a condensation temperature no more $1 \text{ }^\circ\text{C}$ higher than ambient. Therefore, we employed the aluminum radiator with fins (Figure R1) as water condenser, whose heat exchange surface area is 0.247 m^2 , provides enough heat exchange surface to efficiently dissipate the condensation heat release by water vapor. (Page 42)

Additional comments

1. Figure 1: recommend labeling a, b, c... from top to bottom, so the reader can follow the daytime process first, then the nighttime's

Response: Thank you very much for your suggestion!

We think it is more readable and logical to first introduce the working principle of both daytime and nighttime, and then discuss their energy balance and temperature difference. It is clearer to find the difference of working mode and energy analysis between daytime and nighttime. By the way, we improve the quality of Figure 1 according to your suggestions.

2. Figure 5: it could be split into separate figures so each can be enlarged, it is difficult to read legends and see the details

Response: Thank you very much for your suggestion!

We think the subgraphs in Figure 5 are very closely and thus integrated them together in one figure. To make the Figure 5 more clear, we revised the legends and titles, enlarged the size of words in the picture, and adjusted the color according to your suggestions.

Changes add to manuscript:

Figure 5 was revised as following version:

Figure 5. Outdoor Demonstration of the Hybrid SAWH-TEPG Device for Water Production and 24-hour Power Generation.

3. Page 20, line 416: how is the efficiency calculated

Response: Thank you very much for your comment!

The solar-driven water harvesting efficiency was calculated according to the amount of collected water and solar irradiation, as shown in the following equation,

$$\eta = \frac{m_{\text{water}} \Delta H_{\text{water}}}{\int_{t_0}^{t_{\text{end}}} W_{\text{solar}} dt}$$

where the m_{water} represents the mass of harvested water, the ΔH_{water} is the enthalpy (latent heat) of water condensation, and the W_{solar} is the measured real-time solar irradiation intensity during water production process.

Changes add to manuscript:

the solar-driven water harvesting efficiency (η) is calculated as 21.7% by the following equation,

$$\eta = \frac{m_{\text{water}} \Delta H_{\text{water}}}{\int_{t_0}^{t_{\text{end}}} W_{\text{solar}} dt}$$

where the m_{water} represents the mass of harvested water, the ΔH_{water} is the enthalpy (latent heat) of water condensation, and the W_{solar} is the real-time solar irradiation intensity during the water production process. (Page 20)

4. Supplementary figures 1/19/22 are misleading: the thermal network diagrams seem to suggest a heat flow based on temperature levels on a vertical scale, but the temperature locations don't correspond to their levels (e.g., T_{universe} should be all the way to the bottom; it looks like $T_{\text{sor}}=T_{\text{abs}}$ which means there can be no heat flow). I recommend not using this figure in the paper, alternatively, it needs to be revised

Response: Thank you very much for your good suggestion!

We think these thermal network diagrams are valuable to show the energy balance and heat transfer process, and thus we retained these figures. We agree your viewpoint that T_{universe} should be all the way to the bottom and the T_{sor} larger than the T_{abs} . Now, we revised the Supplementary Figures 1/19/22 according to your good suggestions.

Changes add to SI file:

Figure S1, S19, and S22 was revised as following version:

Figure S1. (a) Energy balance model and (b) theoretical thermal equilibrium analysis of the hybrid SAWH-TEPG device during daytime. (c) Energy balance model and (d) theoretical thermal equilibrium analysis of the hybrid SAWH-TEPG device during nighttime.

Figure S19. Comparison of sorption heating power, radiative cooling power, and convective heat loss of the hybrid SAWH-TEPG devices at 22:30 pm and 1:30 am during nighttime.

Figure S22. (a) Theoretical thermal equilibrium results of the hybrid SAWH-TEPG device at (a) 12:30 pm and (b) 14:00 pm during daytime.

References

- [1] Zhang, Y., Ravi, S. K., and Tan, S. C. (2019). Food-derived carbonaceous materials for solar desalination and thermo-electric power generation. *Nano Energy* 65, 104006.
- [2] Zhu, L., Gao, M., Peh, C. K. N., Wang, X., and Ho, G. W. (2018). Self-contained monolithic carbon sponges for solar-driven interfacial water evaporation distillation and electricity generation. *Advanced Energy Materials* 8, 1702149.
- [3] Gao, M., Peh, C. K., Phan, H. T., Zhu, L., and Ho, G. W. (2018). Solar absorber gel: localized macro-nano heat channeling for efficient plasmonic Au nanoflowers photothermal vaporization and triboelectric generation. *Advanced Energy Materials* 8, 1800711.
- [4] Jiang, H., Ai, L., Chen, M., and Jiang, J. (2020). Broadband nickel sulfide/nickel foam-based solar evaporator for highly efficient water purification and electricity generation. *ACS Sustainable Chemistry & Engineering* 8, 10833-10841.
- [5] Zhu, L., Ding, T., Gao, M., Peh, C. K. N., and Ho, G. W. (2019). Shape conformal and thermal insulative organic solar absorber sponge for photothermal water evaporation and thermoelectric power generation. *Advanced Energy Materials* 9, 1900250.
- [6] Zhang, X., Gao, W., Su, X., et al. (2018). Conversion of solar power to chemical energy based on carbon nanoparticle modified photo-thermoelectric generator and electrochemical water splitting system. *Nano Energy* 48, 481-488.

- [7] Zhang, Q., Chen, S., Fu, Z., Yu, H., and Quan, X. (2020). Temperature-difference-induced electricity during solar desalination with bilayer MXene-based monoliths. *Nano Energy* 76, 105060.
- [8] Li, N., Yang, D. J., Shao, Y., et al. (2021). Nanostructured black aluminum prepared by laser direct writing as a high-performance plasmonic absorber for photothermal/electric conversion. *ACS Applied Materials & Interfaces* 13, 4305-4315.
- [9] Liu, X., Mishra, D. D., Li, Y., Gao, L., Peng, H., Zhang, L., and Hu, C. (2021). Biomass-derived carbonaceous materials with multichannel waterways for solar-driven clean water and thermoelectric power generation. *ACS Sustainable Chemistry & Engineering* 9, 4571-4582.
- [10] Cottrill, A. L., Liu, A. T., Kunai, Y., et al. (2018). Ultra-high thermal effusivity materials for resonant ambient thermal energy harvesting. *Nature Communications* 9, 664.
- [11] Yang, K., Pan, T., Pinnau, I., Shi, Z., and Han, Y. (2020). Simultaneous generation of atmospheric water and electricity using a hygroscopic aerogel with fast sorption kinetics. *Nano Energy* 78, 105326.

Finally, we authors thank the editor and reviewers for your good comments and suggestions!

REVIEWERS' COMMENTS

Reviewer #3 (Remarks to the Author):

Very good improvements. I have no further comments.